# SynGAP isoforms differentially regulate synaptic plasticity and dendritic development

Yoichi Araki[1], Ingie Hong[1], Timothy R Gamache[1], Shaowen Ju[1], Leonardo Collado-Torres[2], Joo Heon Shin[2], Richard L Huganir[1]*

[1]Johns Hopkins University School of Medicine, Department of Neuroscience, Kavli Neuroscience Discovery Institute, Baltimore, United States; [2]Lieber Institute for Brain Development, Baltimore, United States

**Abstract** SynGAP is a synaptic Ras GTPase-activating protein (GAP) with four C-terminal splice variants: α1, α2, β, and γ. Although studies have implicated *SYNGAP1* in several cognitive disorders, it is not clear which SynGAP isoforms contribute to disease. Here, we demonstrate that SynGAP isoforms exhibit unique spatiotemporal expression patterns and play distinct roles in neuronal and synaptic development in mouse neurons. SynGAP-α1, which undergoes liquid-liquid phase separation with PSD-95, is highly enriched in synapses and is required for LTP. In contrast, SynGAP-β, which does not bind PSD-95 PDZ domains, is less synaptically targeted and promotes dendritic arborization. A mutation in SynGAP-α1 that disrupts phase separation and synaptic targeting abolishes its ability to regulate plasticity and instead causes it to drive dendritic development like SynGAP-β. These results demonstrate that distinct intrinsic biochemical properties of SynGAP isoforms determine their function, and individual isoforms may differentially contribute to the pathogenesis of *SYNGAP1*-related cognitive disorders.

**\*For correspondence:** rhuganir@jhmi.edu

**Competing interests:** The authors declare that no competing interests exist.

## Introduction

SynGAP is a GTPase-activating protein (GAP) that is highly enriched in dendritic spines of excitatory neurons (*Chen et al., 1998*; *Kim et al., 1998*). SynGAP is a Ras- and Rap- GTPase activating protein that facilitates the hydrolysis of small G protein-bound GTP (active) to GDP (inactive), thus negatively regulating the activity of small G proteins (*Carlisle et al., 2008*; *Chen et al., 1998*; *Pena et al., 2008*; *Rumbaugh et al., 2006*). SynGAP is encoded by the *SYNGAP1* gene and is alternatively spliced to generate 4 distinct C-terminal isoforms: SynGAP-α1, SynGAP-α2, SynGAP-β, and Syn-GAP-γ (*Li et al., 2001*; *McMahon et al., 2012*). The C-terminal domain of SynGAP-α1 contains a class I PDZ ligand sequence (QTRV) which binds MAGUK family proteins such as PSD-95 (*Chen et al., 1998*; *Kim et al., 1998*); (*Grant and O'Dell, 2001*). Heterozygous deletion of *Syngap1* in rodents causes severe deficits in long-term potentiation (LTP) at synapses of hippocampal CA1 pyramidal neurons that are innervated by Schaffer collaterals (SC), as well as severe working memory deficits (*Kim et al., 2003*; *Komiyama et al., 2002*; *Rumbaugh et al., 2006*).

In humans, loss-of-function variants in *SYNGAP1* have been associated with Intellectual Disability (ID), epilepsy, Autism Spectrum Disorders (ASDs), and Neurodevelopmental Disability (NDD). While there are hundreds of genetic risk factors for these disorders, the significantly elevated frequency and 100% penetrance of loss-of-function variants in *SYNGAP1* as well as the range of brain disorders associated with *SYNGAP1* pathogenicity make it unique (*Berryer et al., 2013*; *Carvill et al., 2013*; *Hamdan et al., 2011*; *Hamdan et al., 2009*; *Satterstrom et al., 2020*).

Many loss-of-function variants of the *SYNGAP1* gene have been causally associated with ID, epilepsy, ASD, and other NDDs. In a UK study of 931 children with ID, *SYNGAP1* was the 4th most

highly prevalent NDD-associated gene, and *SYNGAP1* variants accounted for ~0.75% of all NDD cases (*Fitzgerald et al., 2015*). Patients with *SYNGAP1* haploinsufficiency have high rates of comorbid epilepsy, seizures, and acquired microcephaly (*Berryer et al., 2013*; *Carvill et al., 2013*; *Cook, 2011*; *Hamdan et al., 2011*; *Hamdan et al., 2009*; *Parker et al., 2015*; *Rauch et al., 2012*; *Tan et al., 2016*; *Fitzgerald et al., 2015*; *Vissers et al., 2010*; *Vlaskamp et al., 2019*; *Writzl and Knegt, 2013*). Mental Retardation, Autosomal Dominant 5 (MRD5) (OMIM #612621) is caused by mutations in *SYNGAP1*. MRD5 is characterized by moderate-to-severe intellectual disability with delayed psychomotor development apparent in the first years of life (*Holder et al., 2019*). Nearly all reported cases of *SYNGAP1*-related ID and ASD are de novo mutations within/near exons or splice sites of *SYNGAP1* (*Vlaskamp et al., 2019*).

Some key pathophysiological symptoms of ID and ASD observed in *SYNGAP1* patients have been recapitulated in constitutive *Syngap1* hetereozygous (*Syngap1$^{+/-}$*) mice (*Clement et al., 2012*). *Syngap1* heterozygous mice exhibit learning deficits, hyperactivity, and epileptic seizures (*Clement et al., 2012*; *Guo et al., 2009*). Additionally, several MRD5-associated *SYNGAP1* missense mutations also cause SynGAP protein instability (*Berryer et al., 2013*). These data strongly suggest that *SYNGAP1* haploinsufficiency is pathogenic in *SYNGAP1*-associated ID and ASD. Thus, several lines of evidence in mice and humans support that SynGAP is a critical regulator of synaptic plasticity, development, and behavior.

We recently discovered that SynGAP-$\alpha$1 is rapidly dispersed from dendritic spines during LTP, which allows for concomitant spine enlargement and accumulation of synaptic AMPARs (*Araki et al., 2015*). SynGAP-$\alpha$1 dispersion from the dendritic spines releases the inhibition of synaptic RAS activity which is required for the expression of LTP (*Harvey et al., 2008*; *Murakoshi and Yasuda, 2012*; *Walkup et al., 2016*; *Zhu et al., 2002*). Additionally, SynGAP is the third mostly highly expressed protein in the postsynaptic density (PSD) and can undergo multivalent interactions with PSD-95 via liquid-liquid phase separation (LLPS), a process of forming highly concentrated condensates with liquid-like properties, which may contribute to the formation of the PSD complex (*Zeng et al., 2016*). LLPS in cells is a phenomenon in which biochemical reactants are spatially clustered and concentrated in the absence of a surrounding membrane, allowing for organelle-like function without the physical and energetic barriers posed by lipid bilayers (*Shin and Brangwynne, 2017*). Although SynGAP is an ideal candidate to provide the structural basis of PSD (*Zeng et al., 2018*; *Zeng et al., 2016*), the phase separation of SynGAP was extensively characterized only with SynGAP-$\alpha$1. The degree to which the other SynGAP isoforms undergo activity-dependent dispersion and LLPS remains largely unknown, as does the functional significance of these isoforms.

Although *SYNGAP1* haploinsufficiency likely affects the expression of all SynGAP isoforms, only the $\alpha$1 isoform has been rigorously characterized to date. Only a few functional studies of non-$\alpha$1 SynGAP isoforms have been conducted to probe how these isoforms regulate synaptic physiology and disease pathogenesis (*Li et al., 2001*; *McMahon et al., 2012*). In these overexpression studies, the various SynGAP isoforms have been shown to have differing – and even opposing – effects on synaptic transmission (*McMahon et al., 2012*). However, as these were overexpression experiments, endogenous SynGAP was intact in this study, complicating interpretation of these results. It is currently unknown whether *SYNGAP1*-associated ID/ASD pathology is associated with select deficits of specific SynGAP isoforms that may underlie unique features of NDD.

Here, we report that SynGAP-$\alpha$1 constitutes only 25–35% of total SynGAP protein in the brain, underscoring the importance of characterizing how the C-terminal SynGAP splice variants contribute to neuronal and synaptic development that are associated with the pathogenesis of *SYNGAP1* haploinsufficiency. In developing neurons, the various SynGAP isoforms display differences in neuroanatomical and subcellular expression. We report that SynGAP-$\beta$ is expressed earlier in development than the other SynGAP isoforms, and functions specifically to promote dendritic arbor development. In contrast, SynGAP-$\alpha$1 reaches peak expression later in development, and regulates the processes underlying synapse strengthening, including AMPAR insertion and dendritic spine enlargement. Our findings describe unique roles for select SynGAP isoforms in mediating different facets of neuronal function. Furthermore, we identify isoform-specific differences in biochemical interactions between SynGAP and PSD-95, and show how these differences are related to the functional mode of each isoform, regulating either synaptic plasticity or dendritic structure. These results suggest that individual SynGAP isoforms mediate distinct, specialized regulation of neuronal and synaptic development and will inform potential therapeutic strategies for treating *SYNGAP1*-related disorders.

## Results

### SynGAP isoforms have distinct and overlapping expression profiles during brain development

*SYNGAP1* is alternatively spliced at several sites to include exons 18, 19, or 20 to generate four unique C-terminal isoforms: SynGAP-α1, SynGAP-α2, SynGAP-β, and SynGAP-γ (*Figure 1A,B*). Syn-GAP-α1 and SynGAP-α2 isoforms skip exon 19 and are produced by selective splicing of exon 20, whereby SynGAP-α1 contains a PDZ ligand (-QTRV) and SynGAP-α2 lacks this domain. The SynGAP-β isoform includes a frameshifting extension of exon 18 leading to early termination, which generates a SynGAP protein product with a partially truncated coiled-coil domain. The SynGAP-γ isoform includes exon 19, which contains a short coding sequence followed by a STOP codon (-LLIR*).

To characterize each SynGAP isoform, we raised antibodies using SynGAP C-terminal peptides as antigens (*Figure 1B*, black dotted underlines). Antibody specificity was validated in transfected HEK 293 T cells (*Figure 1C* Left 4 lanes, and quantification in *Figure 1—figure supplement 1A*), as well as in brain lysates from WT and *Syngap1* heterozygous (*Syngap1 +/-*) mice, in which immunoblotting demonstrates an expected ~50% reduction of expression of all SynGAP isoforms (*Figure 1C* Right 2 lanes, quantification in *Figure 1—figure supplement 1A*). All four SynGAP isoforms are enriched in brain tissue (* asterisks: non-specific band) with other brain-specific proteins, such as Stargazin and TARP-γ8 (*Figure 1D*, and quantification in *Figure 1—figure supplement 1B*). To determine the expression profile of SynGAP isoforms, we isolated 8 brain regions from adult (P42) mice. All four SynGAP isoforms are enriched in forebrain regions such as the cerebral cortex and hippocampus in comparison to hindbrain structures such as the pons (*Figure 1E*, quantification in *Figure 1—figure supplement 1C*). However, there are several isoform-specific differences in regional expression. For example, SynGAP-β and SynGAP-γ are weakly expressed in the olfactory bulb, and SynGAP-γ is expressed at low levels in the cerebellum. *SYNGAP1* mutations have been linked to NDDs such as ID and ASD, which suggests an important role for *SYNGAP1* in normal brain development. Thus, we sought to investigate the expression of the SynGAP isoforms throughout development in brain tissue from mice at several developmental stages spanning late embryogenesis to adulthood (*Figure 1F–H*, complete set of quantification in *Figure 1—figure supplement 1D*). SynGAP-β is expressed earlier in development (E18-P14) compared to other isoforms, whereas SynGAP-α2 is generally the most abundant isoform and reaches maximal expression at P21-P35. SynGAP-α1 expression also increases later in development (*Figure 1F–H*). Expression of other synaptic proteins (GluA1, PSD-95, and TARPs) reached maximal expression between P21 and P42, which is similar to the timeframe for maximal expression of SynGAP-α1 and SynGAP-α2.

In order to more rigorously quantify the expression levels of SynGAP isoforms over development, using standardized detection ratios of each isoform to total SynGAP based on *Figure 1C*, we calculated the relative abundance (% total SynGAP) of each isoform at P0 and P42 (*Figure 1F–H*). Syn-GAP-β is relatively highly expressed at P0 (34.6 ± 0.6%) and decreases to 15.7 ± 0.8% at P42. SynGAP-α2 expression increases more slowly than the β isoform prenatally, but is also well expressed at P0 (31.9 ± 0.4%). The α2 isoform is the most abundant isoform at P42 (44.9 ± 1.5%), which is consistent with a previous finding that SynGAP-α2 is the dominant isoform at the level of mRNA expression (*Yokoi et al., 2017*). SynGAP-α1 exhibits relatively low expression levels at P0 (24.3 ± 0.3%) and then accumulates throughout development, eventually becoming the second-most highly expressed isoform when measured at P42 (35.0 ± 0.9%) next to the α2 isoform. SynGAP-γ is expressed at low levels throughout development (9.1 ± 0.5% at P0, and 4.3 ± 0.3% at P42) (*Figure 1H*).

To extend our protein-level observations and investigate the correspondence to human *SYNGAP1*, we analyzed previously published human brain RNAseq data (n = 338) (*Jaffe et al., 2018*; *Figure 1I,J*, Reads per 80 million mapped (RP80M) in *Figure 1—figure supplement 1E,F*). Consistent with our biochemical estimates, splice junctions that lead to the β isoform comprised ~22% of all reads spanning the exon 17–18 junction and decreased slightly across development (*Figure 1I*). At the α1/ α2/γ junction, which is relevant in the non-β transcripts (the remaining 78%), we observe that junction reads corresponding to the α2 isoform are the most abundant across all ages (~56% of reads spanning exon 18 to 19/20 junctions), and the α1 isoform follows at ~35%, increasing slowly throughout development, while the γ isoform junction reads are rare (~9% of non-β) (*Figure 1J*). This correspondence with protein data shows that the SynGAP isoform abundance levels are tightly

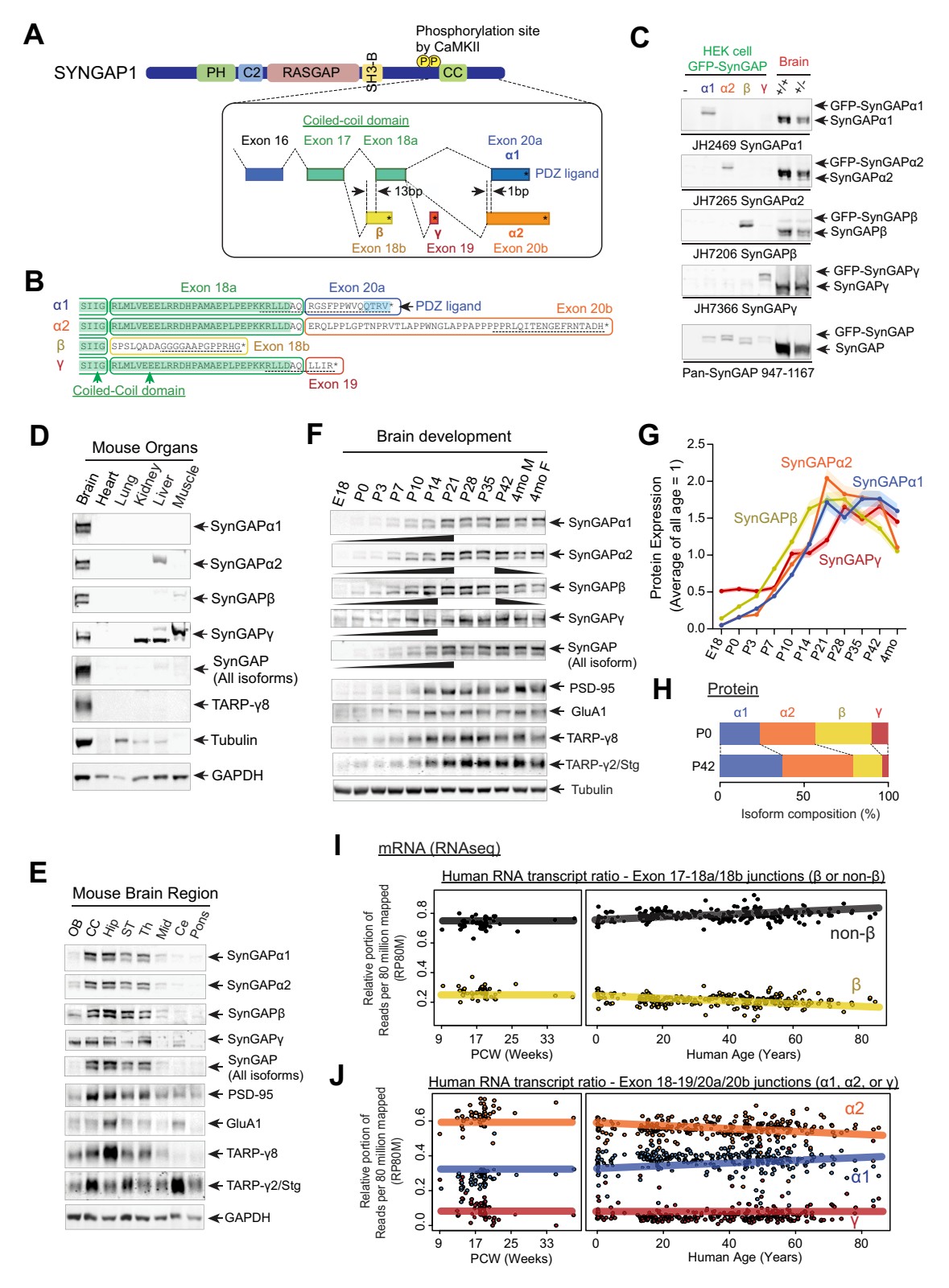

**Figure 1.** SynGAP isoforms are differentially expressed during brain development. (**A**) Schematic of *SYNGAP1* splicing at the C-terminus. *SYNGAP1* is alternatively spliced within exons 18–20 to generate four unique C-terminal isoforms designated as α1, α2, β, and γ. (**B**) C-terminal amino-acid sequences of SynGAP isoforms encoding select protein domains. Coil-Coil domain (yellow) and PDZ ligand-binding domain (blue). Targeted epitopes of isoform-specific SynGAP antibodies (JH2469, JH7265, JH7206, and JH7366) are indicated as dotted lines. (**C**) Specificity of SynGAP isoform-specific

*Figure 1 continued on next page*

*Figure 1 continued*

antibodies. Immunoblots of SynGAP isoform expression in lysates prepared from HEK 293 T cells expressing individual GFP-tagged SynGAP isoforms and lysates prepared from brain tissue obtained from *WT* and *Syngap1* +/- mice were shown. Quantification of relative SynGAP isoform levels with respect to total SynGAP expression measured from immunoblot were shown in *Figure 1—figure supplement 1A*. Two-way ANOVA followed by Tukey's post hoc test (Isoform $F_{(4,30)}$ = 1.900; p=0.13, Genotype $F_{(1,30)}$ = 451.2; p<0.001, Interaction $F_{(4,30)}$=1.900; p=0.13, n = 4 each condition) was performed. Error bar indicates ± SEM. (D) Endogenous expression and distribution of SynGAP isoforms in various organs. Immunoblots of qualitative distribution of SynGAP isoforms in lysates prepared from various organ tissues of WT mice were shown. Asterisks indicate non-specific bands that are also detected in tissue from knockout mice. Two-way ANOVA followed by Tukey's post hoc test (Tissue $F_{(5,144)}$ = 1433; p<0.0001, Isoform $F_{(7,144)}$ = 229.3; p<0.0001, Interaction $F_{(35,144)}$ = 25.45; p<0.0001, n = 4 each condition) was performed. Heat map of immunoblots was displayed in *Figure 1— figure supplement 1B*. The amount of protein in the brain is standardized as 1.0. (E) Western blot of endogenous levels of individual SynGAP isoforms and other synaptic proteins in lysates prepared from several brain regions obtained from *WT* and *Syngap1* +/- mice. (OB: Olfactory bulb, CC: Cerebral cortex, Hip: Hippocampus, ST: Striatum, Th: Thalamus, Mid: Midbrain, Ce: Cerebellum). Two-way ANOVA followed by Tukey's post hoc test (Brain regions $F_{(7, 264)}$=1048; p<0.0001, Molecules $F_{(10,264)}$ = 8.0 x 10$^{-12}$; p>0.9999, Interaction $F_{(70.264)}$ = 59.06; p<0.0001, n = 4 each condition) was performed. Graph showing the mean values of each signal was displayed in *Figure 1—figure supplement 1C*. (F–H) Developmental expression profiles of individual SynGAP isoforms and related synaptic proteins. (F) Immunoblots of SynGAP isoform expression measured in forebrain tissue lysates prepared from *WT* and *Syngap1* +/- mice at different developmental ages. (G) Quantification of immunoblots representing relative enrichments along developmental stage. The mean values of each signal were plotted in the graph. (H) Quantification of absolute SynGAP isoform abundance at P0 and P42 from **C** and **G**. Error bars indicate ± SEM. Two-way ANOVA followed by Tukey's post hoc test (Developmental stage $F_{(10,330)}$ = 397.4; p<0.0001, Molecule $F_{(9,330)}$ = 2.116; p=0.027, Interaction $F_{(90,330)}$ = 26.18; p<0.0001, n = 4 each condition) was performed. (I) mRNA expression of the β and non-β SYNGAP1 isoforms across age in human dorsolateral prefrontal cortex. The relative portion of RNAseq reads spanning the exon 17–18 junction supporting either isoform was plotted against human age (post-conception weeks and years) with a linear regression. (J) mRNA expression of the α1, α2, and γ SYNGAP1 isoforms across age. The relative portion of RNAseq reads spanning the exon 18–19 junction (γ) or 18–20 (α1, α2) junction supporting each isoform was plotted against human age. Reads per 80 million mapped (RP80M) of RNAseq data are shown in *Figure 1—figure supplement 1E,F*.

The online version of this article includes the following figure supplement(s) for figure 1:

**Figure supplement 1.** SynGAP isoforms are differentially expressed during brain development.

---

controlled at the level of splicing, and the relative ratios across development are conserved in mice and humans. Therefore, we next tested whether these isoforms have unique neuronal functions and play distinct roles in *SYNGAP1*-related pathogenesis.

## Unique biochemical properties and subcellular localization patterns of SynGAP isoforms

To better understand potential isoform-specific functions of SynGAP in neurons, we first investigated differences in LLPS, a mechanism for the effective subcellular organization of cellular proteins. We previously discovered that SynGAP-α1 undergoes LLPS with PSD-95 at physiological concentrations in vitro, resulting in the concentration of SynGAP into dense condensates that are reminiscent of the PSD (*Zeng et al., 2016*). To investigate the biochemical and phase separation propensities of other SynGAP isoforms, we first performed a sedimentation assay in HEK 293 T cells transfected with constructs encoding tagged full-length PSD-95 and SynGAP (*Figure 2A*). Here, centrifugation of the sample resulted in two fractions: the insoluble-protein-containing pellet fraction (termed: [P]) and the soluble supernatant fraction (termed: [S]). The ratio of each protein in the condensed phase fraction ([P] /([S] + [P]), termed 'Sedimentation index') was calculated to indicate the propensity of the protein to undergo LLPS. Both myc-PSD-95 and GFP-SynGAP-α1 wild-type (WT) remain mostly in the soluble fraction when expressed alone in HEK 293 T cells (23.1 ± 4.2% of PSD-95 in the pellet fraction, 38.2 ± 0.5% of SynGAP-α1 in the pellet fraction when expressed alone, *Figure 2B*). Co-expression of myc-PSD-95 and GFP-SynGAP-α1 WT causes a dramatic increase in the abundance of both proteins in the pellet [P] fraction (80.3 ± 2.2% of PSD-95 and 74.7 ± 3.3% of SynGAP-α1 in the pellet fraction when co-expressed, ***p<0.001 compared to expressed alone, *Figure 2B*). We previously generated a mutant form of SynGAP-α1 that contains two point mutations – L1202D and K1252D (SynGAP-α1 LDKD) (*Zeng et al., 2016*) – which prevent SynGAP trimerization and phase-separation with PSD-95. First, we tested the effect of the LDKD mutation on the synaptic mobility of SynGAP-α1 by measuring fluorescence recovery after photobleaching (FRAP) of single dendritic spines (*Figure 2—figure supplement 1A*). We found that GFP-tagged full-length SynGAP-α1 LDKD recovers fluorescence in spines to a magnitude greater than that of GFP-SynGAP-α1 WT (*Figure 2— figure supplement 1A*, Recovery plateau for GFP-SynGAP-α1 WT = 0.183 (95% CI = 0.169–0.199);

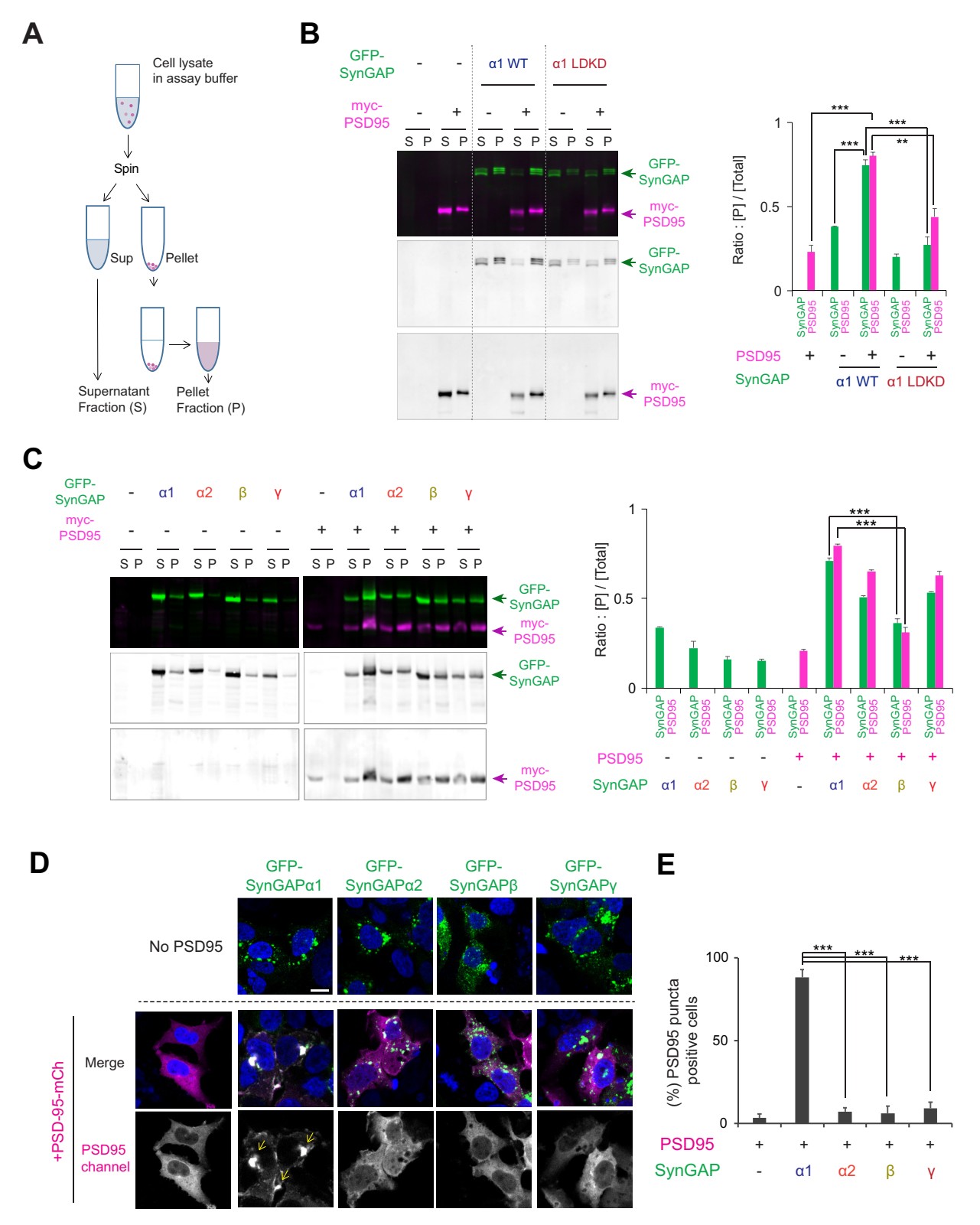

**Figure 2.** Condensation properties and subcellular localization of various SynGAP isoforms in live cells. (**A**) Schematic diagram of LLPS sedimentation assay. HEK 293 T cell lysates were centrifuged and fractionated into insoluble pellet (P) and soluble supernatant (S) fractions. (**B**) Representative immunoblot probing levels of GFP-SynGAP (WT or phase separation mutant) and myc-PSD-95 in phase-separated supernatant and pellet lysate fractions obtained from HEK cells expressing myc-PSD-95 and either GFP-SynGAP-WT or GFP-SynGAP-LDKD constructs. (Right panel) Quantification of

*Figure 2 continued on next page*

*Figure 2 continued*

pellet fraction ratios obtained from averaged immunoblots as the representative example shown in (A). Error bars indicate ± SEM. Two-way ANOVA followed by Tukey's post hoc test (Molecules F (1,30)=3.026; p=0.09, Transfections F(4,30) = 280.7; p<0.0001, Interaction F(4,30)=59.69; p<0.0001, n = 4, \*\*\*p<0.001, \*\*p<0.01, \*<0.05) was performed. (C) Representative western blot probing levels of individual SynGAP isoforms in phase-separated supernatant and pellet lysate fractions obtained from HEK cells expressing myc-PSD-95 and individual GFP-tagged SynGAP isoforms. (Right panel) Quantification of pellet fraction ratios obtained from averaged western blots as the representative example shown in (D). Error bars indicate ± SEM. Two-way ANOVA followed by Tukey's post hoc test (Transfections F(8,54) = 812,2; p<0.0001, Molecules F (1,54)=50.88; p<0.0001, Interaction F(8,54) = 101.5; p<0.0001, n = 4, \*\*\*p<0.001, \*\*p<0.01, \*<0.05) was performed. (D, E) Representative confocal images of living HEK cells expressing myc-PSD-95 alone or myc-PSD-95 and individual SynGAP isoforms. Scale Bar, 10 µm (D). (E) Quantification of the averaged percentage of PSD-95-positive puncta identified in images of living HEK cells as shown in (D). Error bars indicate ± SEM. One-way ANOVA ANOVA followed by Tukey's post hoc test (Transfections F (4, 15)=96.77; p<0.0001, n = 4 independent coverslip, \*\*\*p<0.001, \*\*p<0.01, \*<0.05) was performed.

The online version of this article includes the following figure supplement(s) for figure 2:

**Figure supplement 1.** Condensation properties and subcellular localization of various SynGAP isoforms in live cells.

---

Recovery plateau for GFP-SynGAP-α1 LDKD = 0.372 (95% CI = 0.362–0.384); \*\*\*p<0.0001), indicating that disruption of SynGAP/PSD-95 LLPS without disrupting PDZ-domain-binding results in a measurable decrease in SynGAP PSD association. Previously, we observed that GFP-SynGAP-α1 LDKD displays diminished synaptic localization in neurons when compared to SynGAP-α1 WT (*Zeng et al., 2016*), underscoring the relationship between phase separation and synaptic localization of SynGAP. In biochemical assays, co-sedimentation of GFP-SynGAP-α1 LDKD and PSD-95 was significantly decreased in the [P] fraction when compared to that of GFP-SynGAP-α1 WT and PSD-95 (44.0 ± 5.0% of PSD-95 and 27.3 ± 4.6% of SynGAP-α1 LDKD at condensed phase fraction when SynGAP-α 1 LDKD and PSD-95 was co-expressed, \*\*\*p<0.001 compared to SynGAP-α1 WT and PSD-95 was co-expressed, *Figure 2B*), suggesting that this assay can be used to sensitively probe changes in LLPS propensity, although other factors such as protein aggregation and innate insolubility may still contribute to sedimentation. We also determined that the PDZ ligand (QTRV) of SynGAP-α1, which is required for SynGAP/PSD-95 LLPS in vitro (*Zeng et al., 2016*), is also critical for LLPS-like sedimentation of SynGAP and PSD-95 in this assay (*Figure 2—figure supplement 1B*, Co-sedimentation index: 80.0 ± 3.8% of PSD-95 when co-expressed with SynGAP-α1, but 49.7 ± 2.7% of PSD-95 when co-expressed with SynGAP-α1 ΔQTRV, \*\*p<0.01). These data are consistent with the results of the in vitro cell-free sedimentation assay experiments reported previously (*Zeng et al., 2016*).

We next examined the propensity of each SynGAP isoform to undergo LLPS-like sedimentation with PSD-95 (*Figure 2C*). Expressed singly, all SynGAP isoforms were preferentially found in the soluble fraction. Co-expression of GFP-SynGAP-α1 and myc-PSD-95 dramatically increased the abundance of both proteins in the pellet fraction (71.0 ± 1.4% of SynGAP-α1 in pellet fraction when co-expressed with PSD-95). GFP-SynGAP-α2 and GFP-SynGAP-γ also exhibited enhanced sedimentation in the presence of myc-PSD-95, albeit to a lesser extent than that of GFP-SynGAP-α1 (50.8 ± 0.9% of SynGAP-α2 and 53.3 ± 0.5% of SynGAP-γ in pellet fraction when co-expressed with PSD-95, *Figure 2C*) These isoforms harbor a complete coiled-coil domain but lack the PDZ ligand. In contrast, GFP-SynGAP-β and myc-PSD-95 did not efficiently co-sediment (36.2 ± 2.5% of SynGAP-β in pellet fraction when co-expressed with PSD-95. \*\*\*p<0.001, compared to SynGAP-α1 co-expressed with PSD-95). SynGAP-β lacks the PDZ ligand and contains only a partial coiled-coil domain. These results highlight the necessity and contribution of the coiled-coil domain and PDZ ligand in facilitating interaction between SynGAP and PSD-95.

We next used confocal microscopy to assess SynGAP isoform-dependent biomolecular condensate formation in living HEK 293 T cells (*Figure 2—figure supplement 1C*). We previously reported that GFP-SynGAP-α1 and RFP-PSD-95 undergo LLPS when expressed in living cells, forming liquid-like cytoplasmic droplets (*Zeng et al., 2016*). When expressed alone in HEK 293 T cells, PSD-95-mCherry (PSD-95-mCh) exhibited relatively diffuse cytoplasmic expression (4.8 ± 1.0% of PSD-95 puncta positive cells). In contrast, co-expression of PSD-95-mCh and GFP-SynGAP-α1 WT led to a dramatic increase in distinct cytoplasmic puncta (>1 µm diameter) (85.5 ± 3.3% of PSD-95 puncta positive cells when co-expressed, \*\*\*p<0.001 compared to PSD-95 alone) (*Figure 2—figure supplement 1C*). However, GFP-SynGAP-α1 LDKD did not induce puncta formation when co-expressed with PSD-95-mCh (22.3 ± 5.3% of PSD-95 puncta positive cells, \*\*\*p<0.001 compared to SynGAP-α1 WT and PSD-95 was co-expressed) (*Figure 2—figure supplement 1C*). We next determined the

percentage of cytoplasmic puncta-positive cells following co-expression of PSD-95-mCh along with each SynGAP isoform (*Figure 2D*). While GFP-SynGAP-α1 expression robustly induced the formation of distinct puncta containing PSD-95 (88.2 ± 4.9% of PSD-95 puncta positive cells when SynGAP-α1 and PSD-95 were co-expressed) (*Figure 2D*), PSD-95-containing puncta were largely absent under conditions in which PSD-95-mCh was co-expressed with each of the non-α1 SynGAP isoforms (7.3 ± 2.3%, 6.0 ± 4.3%, 9.0 ± 3.9% when SynGAP-α2, β,γ and PSD-95 were co-expressed respectively, ***p<0.001, compared to SynGAP-α1 and PSD-95 co-expression) (*Figure 2D*). The failure of non-α1 isoforms to induce measurable formation of cytoplasmic puncta suggests that a complete coiled-coil domain and PDZ ligand are necessary for live-cell LLPS of SynGAP in this assay. These results suggest that SynGAP isoforms have unique LLPS properties that are determined by their C-terminal sequences.

Finally, we examined the subcellular distribution patterns of SynGAP isoforms in the mouse brain (*Figure 3A–C*, complete set of quantification in *Figure 3—figure supplement 1*). Mouse brains were excised and fractionated into Total (Total homogenate), S2 (13,800 x g Supernatant), SPM (Synaptosomal plasma membrane), and PSD (Postsynaptic density). Almost all SynGAP isoforms were highly enriched in PSD fractions (α1 7.1 ± 0.5 fold enrichment, α2 6.3 ± 0.2 fold enrichment, β 3.6 ± 0.2 fold enrichment, γ 4.6 ± 0.3 fold enrichment). However, the SynGAP-β isoform was significantly less enriched in PSD (**p<0.001, PSD enrichment of SynGAP-β compared to α1 and α2). Additionally, SynGAP-β was significantly more highly expressed in the S2 (cytosolic) fraction compared to other isoforms. In contrast, the expression of the α1 isoform was very low in this fraction ([S2]*10/[Total] ratio; β 3.3 ± 0.2, ***p<0.001, compared to other isoforms α1 0.11 ± 0.05, α2 0.59 ± 0.11, γ 0.79 ± 0.06). These results indicate that phase separation characteristics of SynGAP isoforms in vitro reflect the subcellular localization patterns of SynGAP isoforms in vivo.

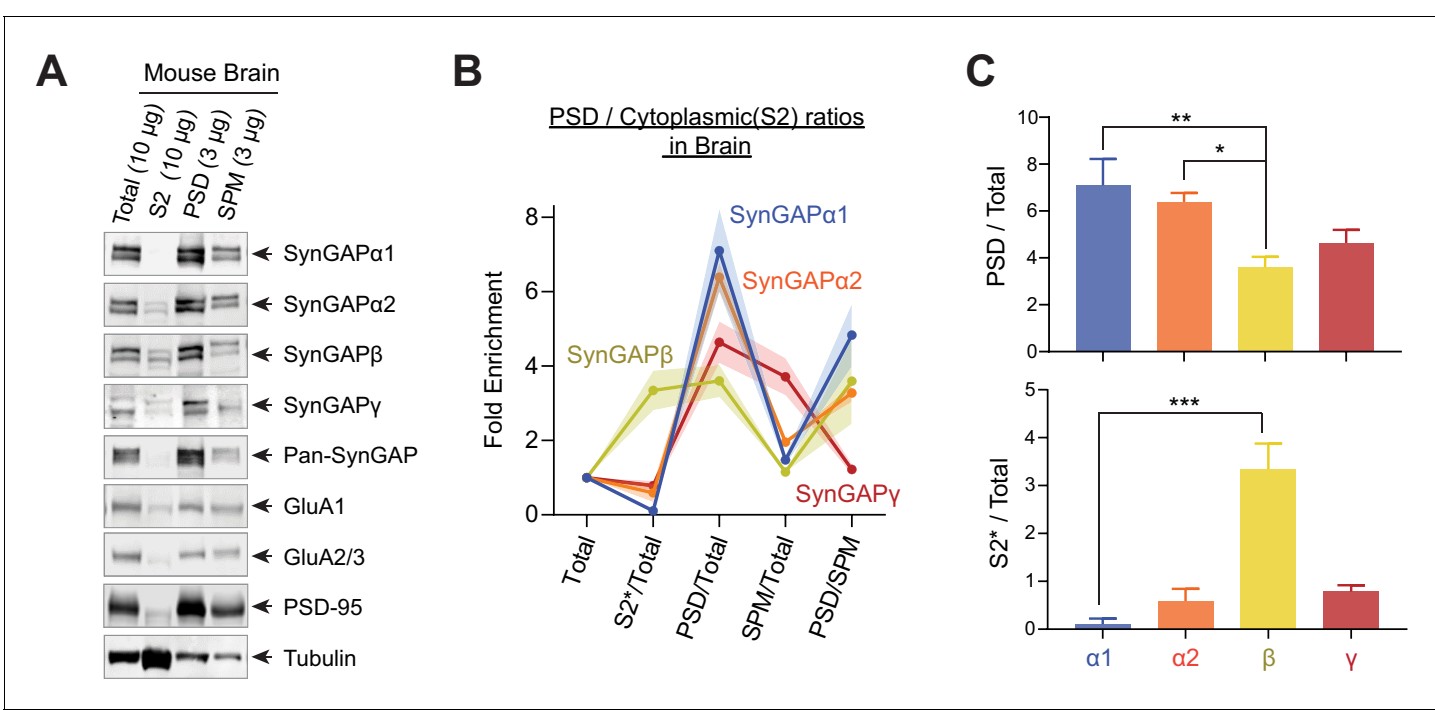

**Figure 3.** Subcellular localization of various SynGAP isoforms in the brain. (**A**) Immunoblot probing endogenous levels of individual SynGAP isoforms and other synaptic proteins in forebrain tissue lysates obtained from adult mice subjected to postsynaptic density fractionation. (**B, C**) Averaged enrichment of SynGAP isoforms in subcellular fractions in comparison to their levels within the total homogenate fraction, S2 fractions, and PSD fractions. Error bars indicate ± SEM. Kruskal-Wallis test followed by Dunn's multiple comparison (PSD: H(4) = 15.98; p=0.0011, S2: H(4) = 18.23, p=0.0004, n = 4–7 independent samples for each molecules, Dunn's multiple comparison ***p<0.001, **p<0.01, *<0.05) was performed.
The online version of this article includes the following figure supplement(s) for figure 3:

**Figure supplement 1.** Subcellular localization of various SynGAP isoforms in the brain.

# SynGAP isoforms differentially regulate GTPase activity to Ras, Rap1, and Rac1

Because the SynGAP isoforms are differentially expressed throughout the brain and display varying ability to associate with synaptic scaffolds, we investigated whether there might also be differences between SynGAP isoforms in their ability to activate RAS family GTPases. To test this possibility, we assayed levels of GTP-bound GTPases such as Ras, Rap1, and Rac1 in HEK 293 T cells expressing several small G-proteins in the presence of individual SynGAP isoforms. Our data demonstrate that specific SynGAP isoforms differentially activate GTP hydrolysis bound to Ras, Rap1, and Rac1 (*Figure 4A–C*). SynGAP-β exhibited the highest GAP activity levels among all isoforms (50.6 ± 3.7% decrease in Ras-GTP, 53.3 ± 6.7% decrease in Rap1-GTP, 39.2 ± 2.7% decrease in Rac1-GTP) (*Figure 4D,E*). SynGAP-α1 preferentially activated GTP-hydrolysis of Ras (28.2 ± 2.8% decrease in Ras-GTP), compared to Rap1 (14.3 ± 4.3% decrease in Rap1-GTP, *p<0.05 compared to Ras) (*Figure 4D,E*). SynGAP-α2 showed a similar trend to decrease Ras-GTP over Rap1-GTP (37.0 ± 3.5%

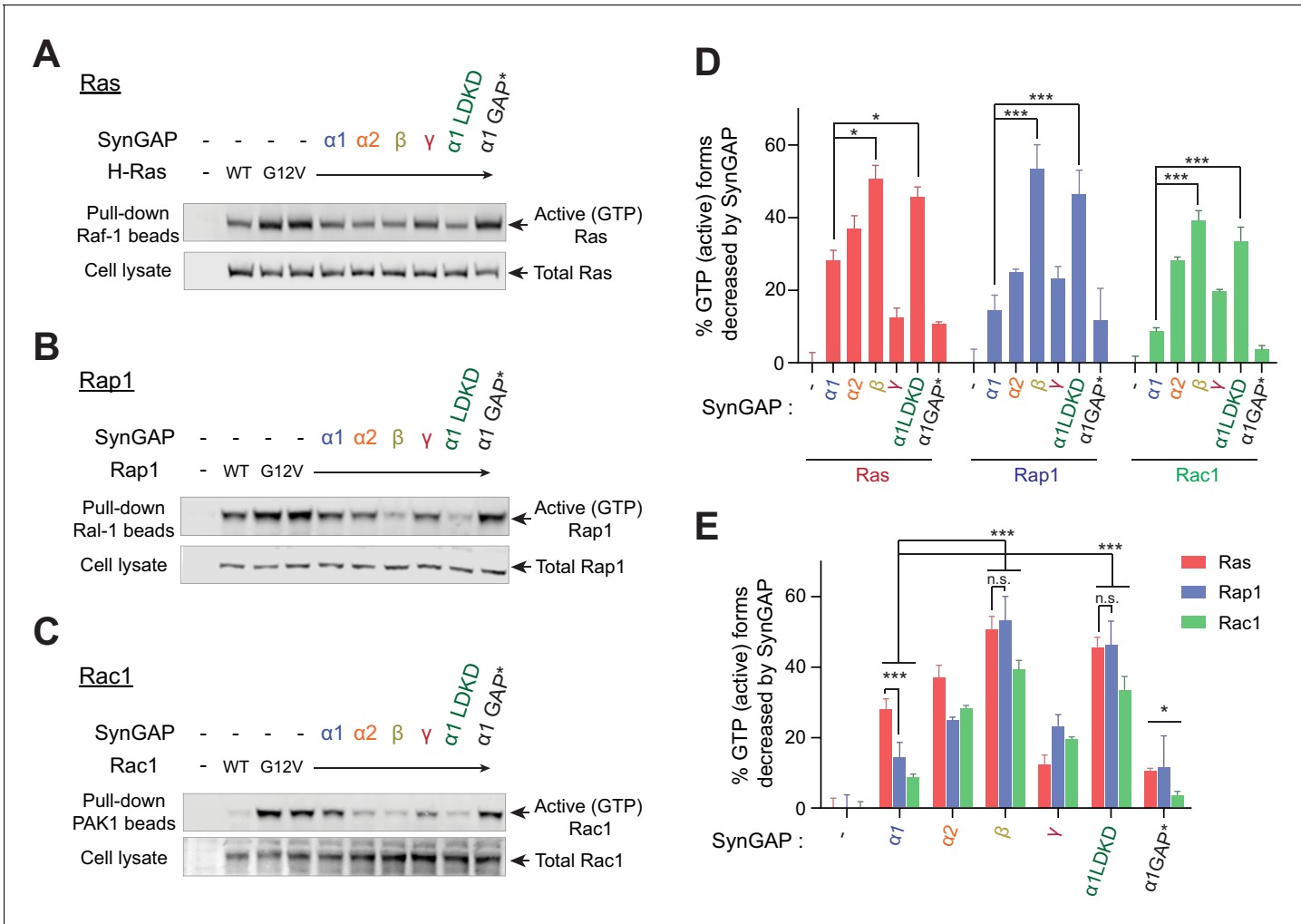

**Figure 4.** SynGAP isoforms differentially regulate the activity of small G proteins. (A) Representative immunoblot detecting levels of active GTP-bound Ras following co-immunoprecipitation of active Ras by pulldown of Raf1 in response to expression of individual SynGAP isoforms in HEK cell lysates. (B) Representative immunoblot detecting levels of active GTP-bound Rap1 following co-immunoprecipitation of active Rap1 by pulldown of Ral1 in response to expression of individual SynGAP isoforms in HEK cell lysates. (C) Representative immunoblot detecting levels of active GTP-bound Rac1 following co-immunoprecipitation of active Rac1 by pulldown of PAK1 in response to expression of individual SynGAP isoforms in HEK cell lysates. (D, E) Quantification of averaged percent reduction of active GTP-bound forms of Ras, Rap1, and Rac1 normalized to total (active + inactive) levels in response to expression of individual SynGAP isoforms expressed in HEK cell lysates. Error bars indicate ± SEM. Two-way ANOVA followed by Tukey's post hoc test (SynGAP isoforms $F_{(6,105)}$ = 62.76; p<0.0001, small G proteins $F_{(2,105)}$=7.414; p=0.0010, Interaction $F_{(12,105)}$ = 2.207; p=0.016, n = 6, ***p<0.001, **p<0.01, *<0.05) was performed.

decrease in Ras-GTP, compared to 24.8 ± 1.0% decreases in Rap1-GTP, p=0.06). Conversely, Syn-GAP-β robustly activated GTP hydrolysis of Rap1 and Ras to similar levels (53.2 ± 6.8% decreases in Rap1-GTP, compared to 50.6 ± 3.8% decreases in Ras-GTP) (*Figure 4D,E*). Since small G proteins were not phase-separated, the soluble SynGAP-β isoform may have greater exposure to cytosolic small G-proteins than do the other less-soluble isoforms, and thus, demonstrates the highest GAP activity levels. However, it is also possible that different C-terminal structures enhance or diminish the accessibility of various small G proteins to the GAP domain and thus differentially regulate their GAP activity.

## Differential dispersion dynamics of SynGAP isoforms during LTP

Previously, we have shown that SynGAP-α1 undergoes rapid NMDAR-CaMKII-dependent dispersion from the synapse, which is required for AMPAR insertion and spine enlargement during LTP (*Araki et al., 2015*). In order to investigate the dispersion dynamics of the other SynGAP isoforms during LTP, we employed a knockdown-replacement strategy in cultured hippocampal neurons, whereby endogenous SynGAP expression was depleted via shRNA-mediated knockdown and individual GFP-tagged, shRNA-resistant SynGAP isoforms were transfected (*Figure 5*). We knocked down 77.3%±0.1% of endogenous SynGAP by shRNA and replaced with similar amount (−90% of endogenous proteins) by shRNA resistant SynGAP isoform construct (*Figure 5—figure supplement 1*). Cultured neurons were subjected to a chemical LTP (chemLTP) treatment during live confocal imaging, and the amount of synaptically localized GFP-tagged SynGAP was measured along with dendritic spine size before and after LTP (*Figure 5A,B*). In this chemLTP stimulation, the magnesium in the media was withdrawn in conjunction with glycine perfusion. With spontaneous glutamate release from axonal terminals, glycine strongly and specifically stimulates synaptic NMDA receptors (*Liao et al., 2001*; *Lu et al., 2001*). GFP-SynGAP-α1 exhibited high synaptic localization prior to LTP induction and then underwent rapid dispersion following LTP (3.5 ± 1.3 fold synaptic spine enrichment of SynGAP-α1 before chemLTP, 1.7 ± 0.3 fold synaptic spine enrichment after chemLTP, ***p<0.001) (*Figure 5A,B*). GFP-SynGAP-α2 was also dispersed albeit to a lesser extent than GFP-SynGAP-α1 (2.8 ± 0.6 fold synaptic spine enrichment of SynGAP-α2 before chemLTP, 1.8 ± 0.4 fold synaptic spine enrichment after chemLTP, *p<0.05). In contrast, GFP-SynGAP-β was less enriched at synapses and failed to disperse upon chemLTP stimulation (1.9 ± 0.09 fold synaptic spine enrichment of SynGAP-α2 before chemLTP, 1.4 ± 0.11 fold synaptic spine enrichment after chemLTP, not significant p>0.05). (*Figure 5A,B*). These data demonstrate dramatic differences in chemLTP-dependent synaptic dispersion dynamics between individual SynGAP isoforms, and indicate a potential role for isoform-specific effects on neuronal and synaptic function.

## Synaptic AMPAR insertion and spine enlargement during LTP are regulated primarily by SynGAP-α1

We previously demonstrated that SynGAP-α1 undergoes rapid NMDAR- and CaMKII-dependent dispersion from the synapse, and this dispersion is required for synaptic AMPAR insertion and spine enlargement that occur during LTP (*Araki et al., 2015*). So far, we have determined that the various SynGAP isoforms differ in their LLPS propensity, GAP activity, localization, and dispersion kinetics during LTP. Thus, we hypothesize that these SynGAP isoforms function differentially during LTP. To test this hypothesis in cultured neurons, we replaced endogenous SynGAP with an shRNA-resistant form of one SynGAP isoform tagged with Azurite (*Araki et al., 2015*; *Zeng et al., 2016*). We also transfected these neurons with the pH-sensitive super-ecliptic-pHluorin-tagged-GluA1 (SEP-GluA1) and mCherry to monitor surface AMPAR expression and dendritic spine size, respectively, in response to chemLTP treatment (*Figure 6A–E*; *Lin et al., 2009*). Under control conditions, significant increases in synaptic-membrane-localized AMPARs and dendritic spine size were observed following LTP stimulation (2.5 ± 1.2 fold synaptic enrichment of AMPA receptor in synaptic spines [***p<0.001] and 2.7 ± 1.4 fold synaptic spine size [***p<0.001] after chemLTP compared to basal condition) (*Figure 6B* and *Figure 6C–E*). Dendritic spine enlargement and synaptic AMPAR accumulation at synapses were occluded when endogenous SynGAP expression was depleted via shRNA-mediated knockdown; this is due to elevated Ras activity, spine enlargement and synaptic AMPAR accumulation in unstimulated baseline conditions (*Araki et al., 2015*) (2.0 ± 1.2 fold enrichment of AMPA receptor in basal state to 2.3 ± 0.9 fold enrichment after chemLTP [Not significant, p>0.05]/

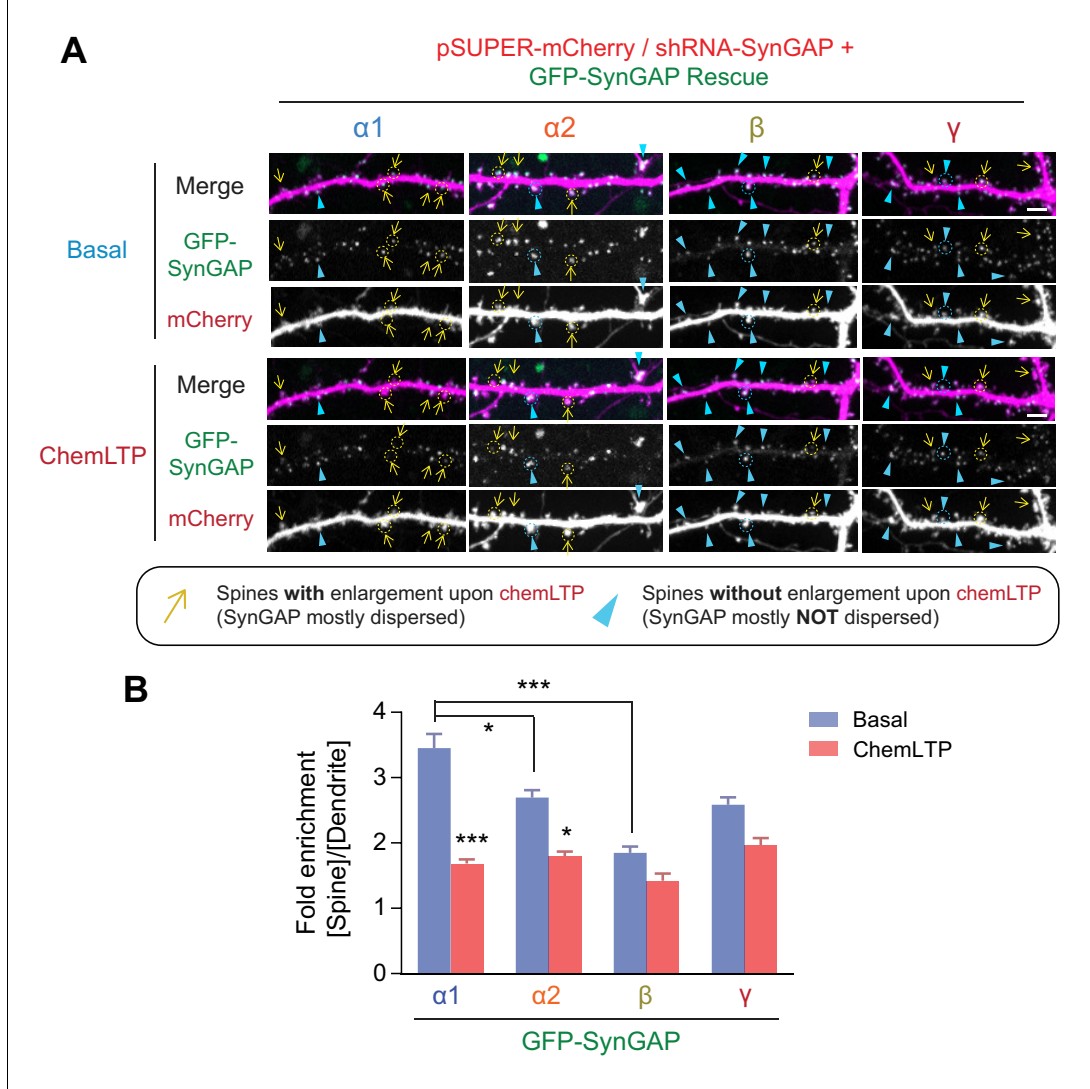

**Figure 5.** Dispersion dynamics of various SynGAP isoforms from synaptic spines during LTP. (**A**) Live confocal images of hippocampal neurons expressing individual GFP-tagged SynGAP isoforms and mCherry during basal conditions and chemLTP conditions. Yellow arrows indicate dendritic spines that enlarged following chemical LTP treatment. Blue arrowheads mark dendritic spines that did not enlarge after chemical LTP treatment. Scale Bar, 5 μm. (**B**) Quantification of averaged relative change in the GFP-SynGAP isoform at synaptic spines following chemLTP treatment. Synaptic localization of SynGAP isoforms was determined by calculating the ratio of GFP intensity within dendritic spine heads and dividing by GFP intensity localized to the dendritic shaft at the base of the dendritic spine. Error bars indicate ± SEM. Two-way ANOVA followed by Tukey's post hoc test (SynGAP isoforms $F_{(3,296)}$ = 21.43; p<0.0001, chemLTP $F_{(1,296)}$=119.9; p<0.0001, Interaction $F_{(3,296)}$ = 6.607; p<0.0001, n = 37–39 spines from 4 independent experiments, ***p<0.001, **p<0.01, *<0.05) was performed. Error bar indicates ± SEM.

The online version of this article includes the following figure supplement(s) for figure 5:

**Figure supplement 1.** shRNA efficiency and rescue SynGAP construct titration assay.

2.1 ± 1.6 fold synaptic spine size in basal state to 2.5 ± 1.6 fold spine size after chemLTP [Not significant, p>0.05] in *SYNGAP1*-shRNA) (*Figure 6A1* and *Figure 6C–E*). Molecular replacement with SynGAP-α1 restored baseline SEP-GluA1 and mCherry intensities to levels comparable to those measured in baseline control conditions and rescued LTP-dependent enhancement of dendritic spine volume and surface AMPAR content (1.1 ± 0.7 fold enrichment of AMPA receptor in basal state to 2.3 ± 0.6 fold enrichment after chemLTP [***p<0.001]/1.4 ± 0.8 fold synaptic spine size in basal state become 2.4 ± 0.7 fold spine size after chemLTP [***p<0.001] in *SYNGAP1*-shRNA + SynGAP-α1 expression) (*Figure 6A2* and *Figure 6C–E*). SynGAP-α2 underwent modest dispersion following stimulation and rescued basal spine enlargement and AMPAR insertion after chemLTP to a

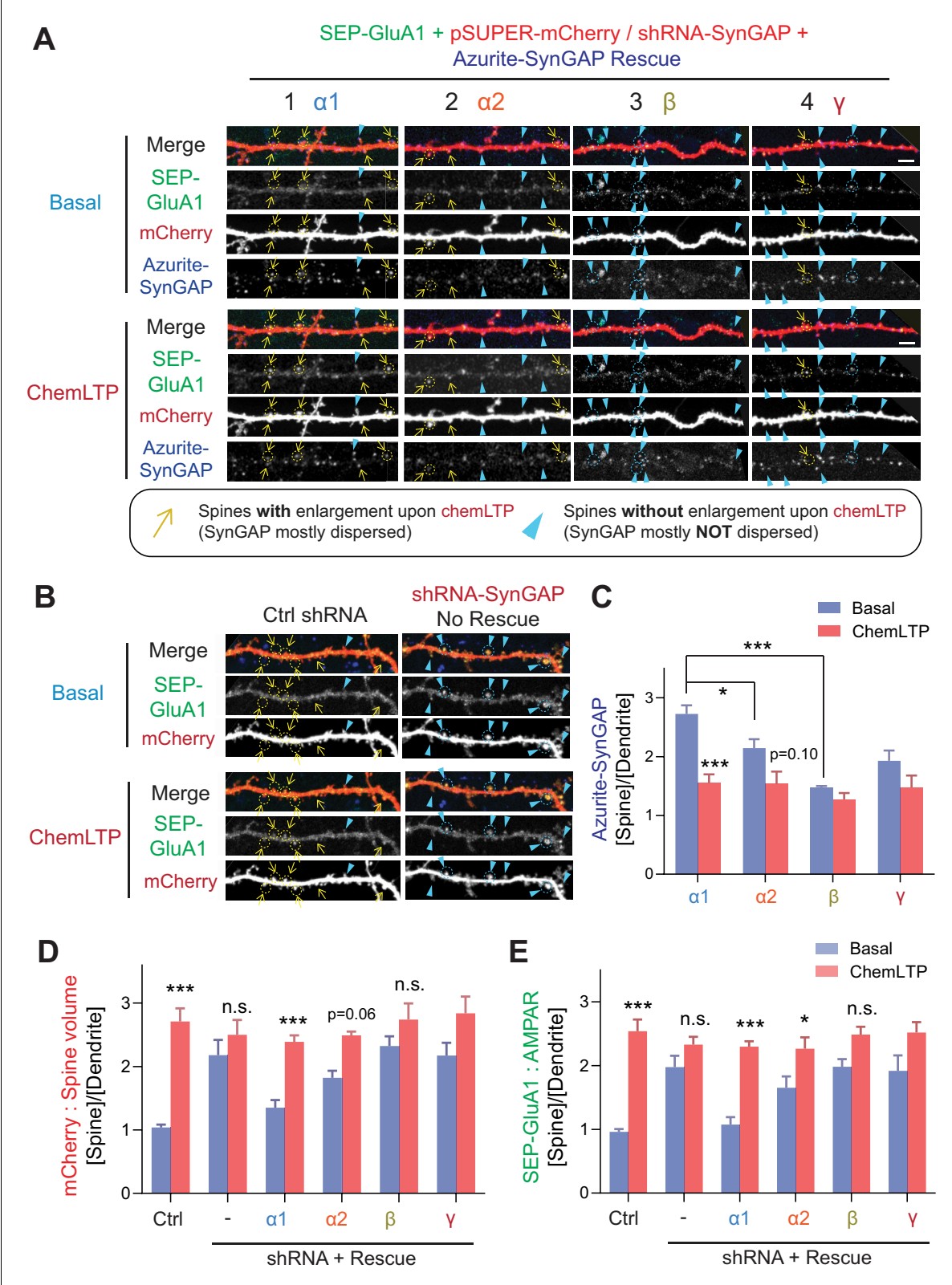

**Figure 6.** SynGAP-α1 rescues AMPA receptor trafficking and structural plasticity deficits in SynGAP-depleted hippocampal neurons. (A, B) Representative live confocal images of cultured hippocampal neurons expressing SEP-GluA1, mCherry, and individual Azurite-tagged SynGAP isoforms in basal and chemLTP conditions. Endogenous SynGAP was knocked-down and replaced with individual Azurite-tagged SynGAP isoforms. Yellow arrows indicate dendritic spines that exhibited LTP-induced enlargement. Scale Bar, 5 μm. (C–E) Quantification of averaged SEP-GluA1, mCherry,

*Figure 6 continued on next page*

*Figure 6 continued*

Azurite-SynGAP intensity in dendritic spines on hippocampal neurons expressing individual SynGAP isoforms before and after chemLTP treatment. Error bars indicate ± SEM. Two-way ANOVA followed by Tukey's post hoc test (ChemLTP F(1,242) = 501.1 [GluA1], 426.4 [mCherry], 219.4 [SynGAP], p<0.001; Genotype F(5,242) = 30.68 [GluA1], 35.71 [mCherry], 553.7 [SynGAP], p<0.001; Interaction (5,240)=15.02 [GluA1], 18.57 [mCherry], 553.7 [SynGAP], p<0.001; n = 47–49 spines from 4 independent coverslips each condition, ***p<0.001, **p<0.01, *<0.05) was performed. Error bar indicates ± SEM.

much lesser extent than SynGAP-α1 (1.7 ± 1.3 fold enrichment of AMPA receptor in basal state to 2.3 ± 1.2 fold enrichment after chemLTP [*p<0.05]/1.8 ± 0.8 fold synaptic spine size in basal state to 2.5 ± 0.4 fold spine size after chemLTP [Not significant p=0.06] in *SYNGAP1*-shRNA + SynGAP-α2 expression) (*Figure 6A3* and *Figure 6C–E*). Replacement with SynGAP-β failed to rescue basal spine enlargement and AMPAR insertion following chemLTP treatment (2.0 ± 0.8 fold enrichment of AMPA receptor in basal state to 2.5 ± 0.8 fold enrichment after chemLTP [Not significant, p>0.05]/ 2.4 ± 1.1 fold synaptic spine size in basal state to 2.8 ± 1.8 fold spine size after chemLTP [Not significant, p>0.05] in *SYNGAP1*-shRNA + SynGAP-β expression) (*Figure 6A4* and *Figure 6C–E*). We previously found that the phase separation mutant of SynGAP-α1 (LDKD) only partially rescued the LTP and significantly lowered the LTP threshold (*Zeng et al., 2016*). These results suggest that both the coiled-coil domain and PDZ ligand are required for LTP rescue in SynGAP KD neurons, and only SynGAP-α1 harbors the necessary and sufficient domains for efficient LTP expression. Our data suggest a specialized role for SynGAP-α1 in regulating LTP.

## Dendritic arbor development is regulated predominantly by SynGAP- β

Since our data suggest that non-α1 isoforms only modestly regulate synaptic plasticity despite their confirmed ability to regulate G-protein activity, we decided to investigate whether these isoforms are involved in other aspects of neuronal function. A previous report showed that *Syngap1* +/- mice exhibit dysregulated dendritic arbor development (*Aceti et al., 2015*). Thus, we assessed the effects of specific SynGAP isoforms in regulating dendritic development. In this experiment, the effects of *SYNGAP1* knockdown on dendritic branching was assessed by comparing control and *SYNGAP1* shRNA-expressing hippocampal neurons at DIV 8 (*Figure 7A*). Obvious basal (< 10-50 μm in length) dendrites (Control: 3.4 ± 0.5 intersections at 10 μm) and a branched primary apical (>100-150 μm in length) dendrite emanate from the somas of control neurons (Control: 4.2 ± 0.3 intersections at 150 μm). Sholl analysis revealed that *SYNGAP1* knockdown aberrantly enhances the number of neurite extensions proximal to neuronal cell bodies (8.3 ± 0.8 interactions at 10 μm, *** p < 0.001 compared to Control 3.4 ± 0.5 intersections at 10 μm) (*Figure 7B, F*). In contrast, *SYNGAP1* knockdown significantly decreased distal branches (*SYNGAP1*-shRNA: 0.3 ± 0.2 intersections at 150 μm, *** p < 0.001 compared to Control: 4.2 ± 0.3 intersections at 150 μm) (*Figure 7B, F*). The aberrantly elevated outgrowth of neurites proximal to neuron somas that is associated with *SYNGAP1* knockdown was successfully rescued by overexpression of each SynGAP isoform (α1 rescue: 4.0 ± 0.5 intersections at 10 μm, α2 rescue: 3.0 ± 0.3 intersections at 10 μm, β rescue: 3.7 ± 0.5 intersections at 10 μm, γ rescue 4.3 ± 0.2 intersections at 10 μm, *** p < 0.001 compared to *SYNGAP1*-shRNA: 8.3 ± 0.8 interactions at 10 μm) (*Figure 7C, G*). Interestingly, only SynGAP-β effectively rescued the distal dendritic complexity deficits (150 μm) by restoring the formation of primary and secondary apical dendrites (β rescue: 3.8 ± 0.6 intersections at 150 μm *** p < 0.001 compared to *SYNGAP1*-shRNA 0.3 ± 0.2 intersections at 150 μm). All other isoforms (α1, α2, and γ) failed to rescue distal branching deficits (α1 rescue: 1.7 ± 0.4 intersections at 150 μm, α2 rescue: 1.3 ± 0.2 intersections at 150 μm, γ rescue: 1.0 ± 0.3 intersections at 150 μm, not significant compared to *SYNGAP1*-shRNA: 0.3 ± 0.2 interactions at 150 μm) (*Figure 7C, G*). Interestingly, expression of SynGAP-α1 LDKD rescued the primary dendrite phenotype, similar to the effect of expression of SynGAP-β (α1 LDKD rescue: 3.5 ± 0.4 intersections at 150 μm *** p < 0.001, β rescue: 3.8 ± 0.6 intersections at 150 μm ***p < 0.001, compared to *SYNGAP1*-shRNA: 0.3 ± 0.2 intersections at 150 μm) (*Figure 7C, D, H*). This result suggests that disruption of SynGAP-α1 LLPS results in more β-like function, rescuing distal dendritic arbor deficits despite containing an α1 C-terminus. Finally, we found that a GAP mutant of SynGAP (*Araki et al., 2015*) does not rescue the dendritic arbor deficits (α1 GAP* rescue: 7.7 ± 1.2 intersections at 10 μm, not significant compared to *SYNGAP1*-shRNA: 8.3 ± 0.8 interactions at 10 μm),

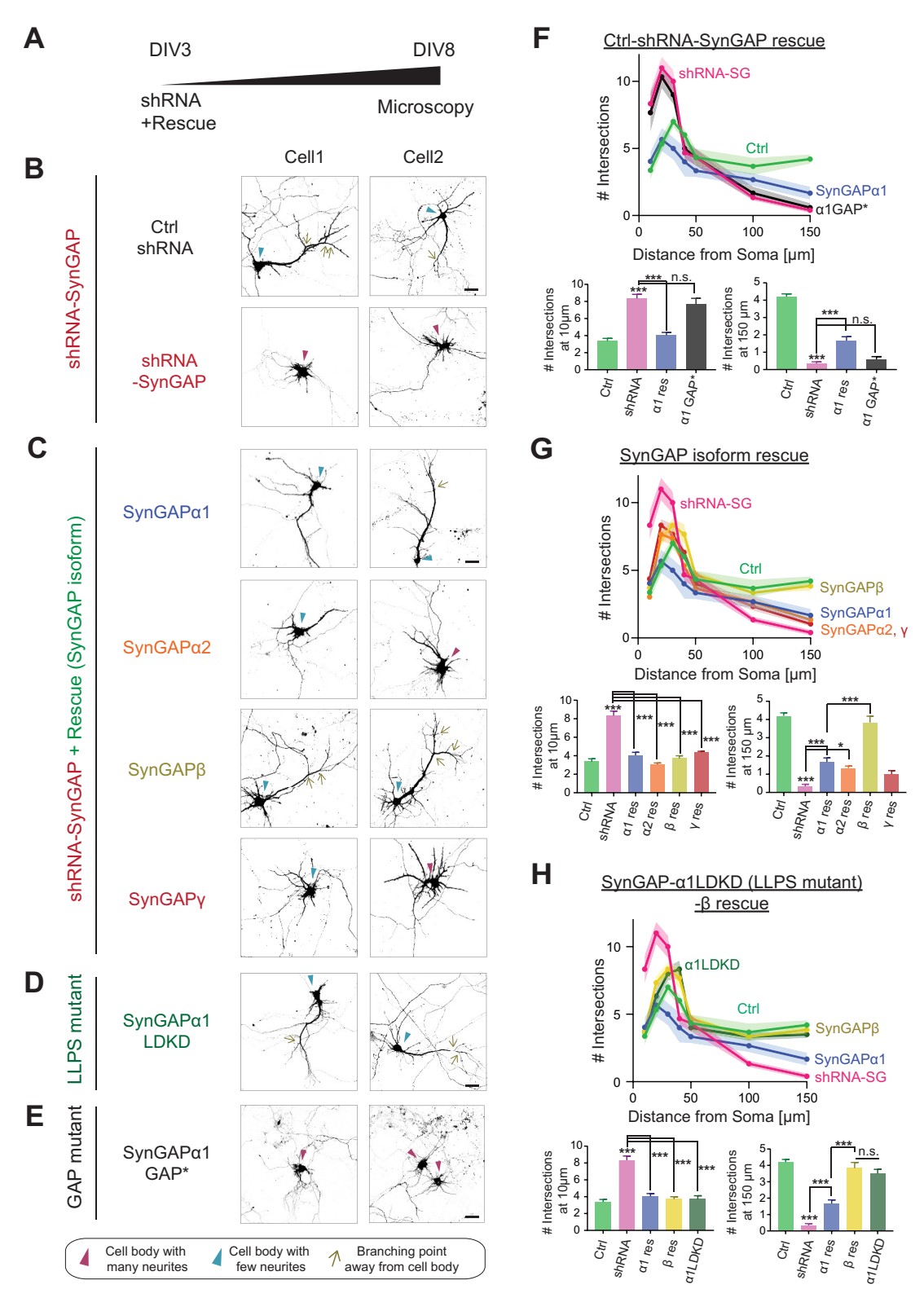

**Figure 7.** SynGAP-β rescues aberrant dendritic arbor development in SynGAP-depleted neurons. (**A**) Schematic of experimental timeline for assessing the effects of individual SynGAP isoform expression on dendritic development. Neuronal morphology was evaluated by observing co-transfected DsRed at DIV 8. (**B-E**) Representative images of dendritic arbors of young cultured hippocampal neurons expressing DsRed upon SynGAP knockdown (**B**) and upon *SYNGAP1* knockdown plus expressing individual SynGAP isoforms (**C**), SynGAPα1 LLPS mutant (**D**), or SynGAPα1 GAP mutant (**E**). Scale

*Figure 7 continued on next page*

*Figure 7 continued*

Bar, 20 μm. (F–H) Sholl analysis of dendritic branches presented as the mean number of intersections plotted as a function of distance from the center of the cell body (center = 0). Error bars indicate ± SEM. Two-way ANOVA followed by Tukey's post hoc test (Distance $F_{(6,952)}$ = 288.6, p<0.001; Genotype $F_{(7,952)}$ = 21.96, p<0.001; Interaction $(42,952)$=14.83, p<0.001, n = 18) was performed. Error bar indicates ± SEM.

indicating that GAP activity is required for the dendritic phenotype rescue described here (*Figure 7E, F*).

## Discussion

Here, we have characterized the developmental expression and subcellular localization, biochemical properties, and functional roles of each C-terminal SynGAP splice variant. Our data suggest novel roles for SynGAP isoforms in regulating dendrite and synapse development in neurons. SynGAP-β is expressed at higher levels than the other isoforms early in brain development, and is gradually replaced in the mature brain through an increase in the expression of SynGAP-α1 and SynGAP-α2. Although SynGAP-β appears dispensable for synaptic plasticity function, it exhibits the strongest GAP activity of all the C-terminal isoforms, preferentially targeting Rap1 and facilitating dendritic arbor development. SynGAP-β not only lacks a PDZ ligand, but also lacks a full coiled-coil domain, likely leading to the dramatically increased cytoplasmic localization. SynGAP-α1, however, contains a complete coiled-coil domain and PDZ ligand, allowing for LLPS with PSD-95 (*Zeng et al., 2016*) and robust concentration at the PSD. We find SynGAP-α1 to be uniquely critical for LTP expression. This dense packing in the PSD in turn may be important to allow the dynamic dispersion of SynGAP during LTP (*Figure 8A*), which we have shown previously to be required for both spine growth and AMPAR trafficking (*Araki et al., 2015*). Our data suggest that while SynGAP is important for the regulation of both plasticity and dendritic development, the biophysical and localization properties of the various isoforms are closely related to their functional role: isoforms that undergo LLPS and cluster densely in the PSD regulate synaptic plasticity and function while those that express more cytoplasmically dominate the regulation of dendritic arbor development. This finding is strengthened by our observation that disruption of SynGAP-α1 LLPS (through the LDKD mutation) resulted in a switch from regulation of synaptic plasticity to regulation of dendritic development in a manner similar to SynGAP-β, despite the fact that the C-terminus retains the SynGAP-α1 PDZ ligand.

### SynGAP-β: early expression and strong GAP function - Roles in dendritic development and their implications for neurodevelopmental disorders

In the present study, we discovered that knockdown of SynGAP results in excessive proximal dendritic sprouting in immature hippocampal neurons. Importantly, only SynGAP-β effectively rescues this developmental phenotype (*Figure 7*). Various small G proteins such as Ras, Rap1, Rac1, and RhoA tightly regulate dendritic arbor development by precisely controlling the number and length of dendritic branches (*Fu et al., 2007*; *Nakayama et al., 2000*; *Saito et al., 2009*; *Sepulveda et al., 2010*). For example, Rap1 increases proximal dendritic branching in rat cortical neurons, and Rap2 activation decreases the length and complexity of developing axonal and dendritic branches (*Chen et al., 2005*). These data link our observed dendritic phenotype caused by SYNGAP1 deficiency to regulation of small G-proteins. Further, dominant-negative forms of Rac1 decrease proximal dendritic branching and increase distal dendritic branching in hippocampal organotypic slice cultures, suggesting that a proper balance of small G-protein activation is crucial for normal dendritic development (*Nakayama et al., 2000*). Additionally, the Ras-PI3K–Akt–mTOR pathway controls somatic and dendritic sizes and coordinates with Ras-mitogen-activated protein kinase signaling to maintain dendritic complexity (*Kumar et al., 2005*). Thus, it is possible that SYNGAP1 haploinsufficiency causes overactivation of Ras and Rap1 and consequently disrupts the balanced signaling required for normal dendritic development. Our results suggest that SynGAP-β plays pivotal role in establishing this balance by regulating Rap1 and other small G proteins. There are currently a variety of available downstream inhibitors of small G proteins, which may prove to be valuable therapeutic targets to ameliorate dendritic deficits caused by SynGAP-β deficiency.

Recent studies have suggested that human induced pluripotent stem cells (hiPSCs) derived from ASD patients exhibit accelerated dendritic outgrowth and excessive dendritic branching following

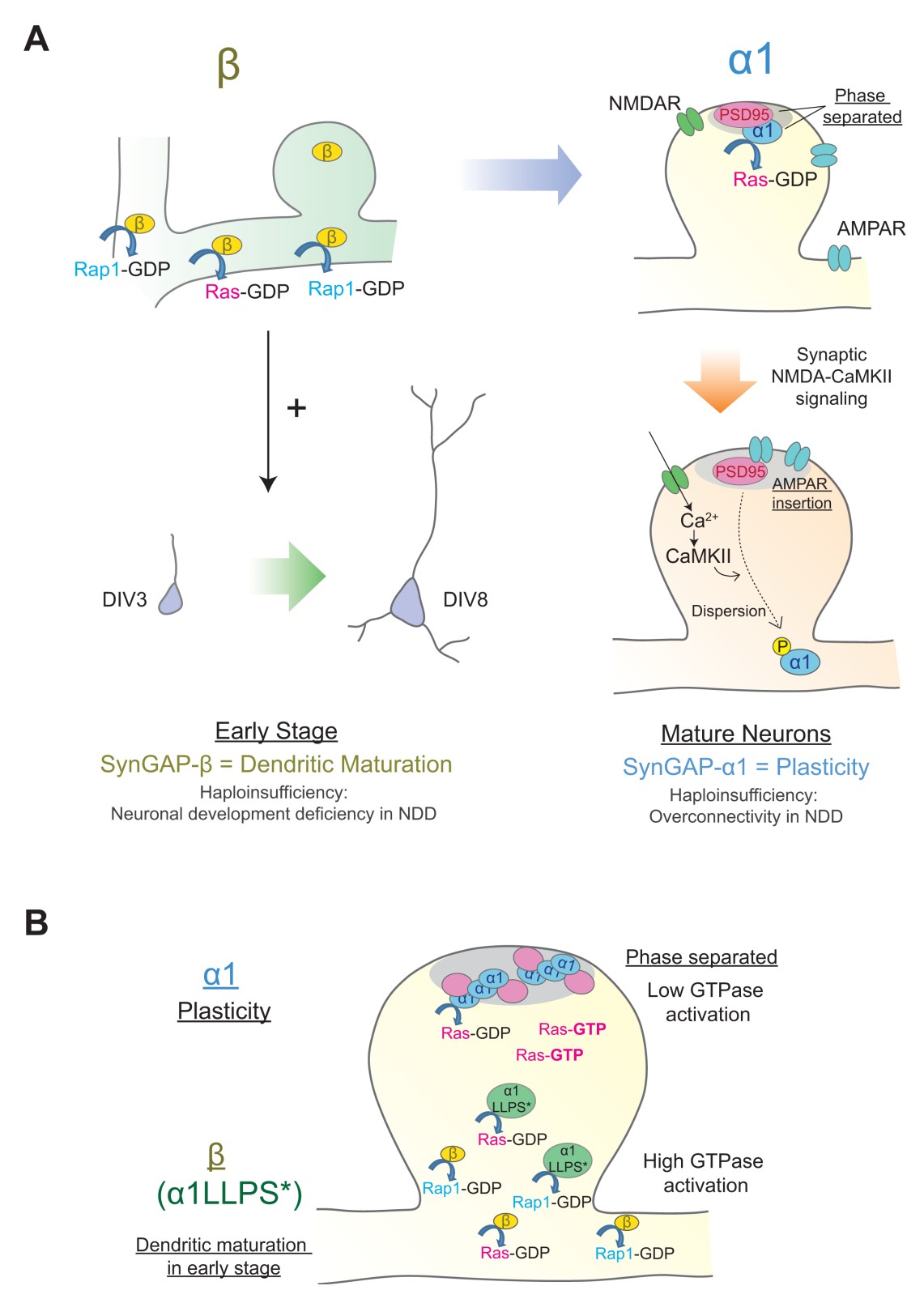

**Figure 8.** Distinct roles of individual SynGAP isoforms in neuronal development and synaptic plasticity. (**A**) Schematics illustrating isoform-specific roles for SynGAP in neuronal maturation and synaptic plasticity. SynGAP-β is expressed early in development, has the lowest LLPS propensity resulting in cytosolic localization, possesses the highest GAP activity in cells, and promotes normal dendritic development. SynGAP-β deficiency may be relevant to the neuronal development deficits in neurodevelopmental disorders (NDD). SynGAP-α1 is expressed later in development, undergoes strongest LLPS

*Figure 8 continued on next page*

Figure 8 continued

in spines resulting in dense expression in the PSD at the basal state, and is rapidly dispersed upon synaptic NMDAR-CaMKII activation. SynGAP-α1 deficiency may be relevant to the synaptic plasticity deficits and overconnectivity in NDD. (B) Schematics illustrating the phase-separation and the localization/functions of SynGAP isoforms. SynGAP-α1 is tightly packed at PSD by phase separation and has the ability of low GTPase activations. In contrast, SynGAP-β is less phase-separated and localized more in cytoplasmic region in synapses and dendritic shafts. It has a strong ability of GTPase activation. The phase-separation mutant of SynGAP-α1 LLPS*) behaves similarly to SynGAP-β.

neuronal differentiation (*Schafer et al., 2019*). These observations, together with our finding that SynGAP-β predominantly promotes dendritic arbor development, links dendritic morphological deficits to the abnormal neuronal wiring associated with NDDs.

## Critical role of SynGAP-α1 in synaptic plasticity; strong interaction with PSD-95 for synaptic enrichment and dispersion during LTP

We have previously reported that SynGAP-α1 is rapidly dispersed in response to LTP-inducing synaptic activity. This dispersion allows for AMPAR insertion into the synaptic membrane and for enlargement of dendritic spines. Thus, SynGAP-α1 functions to regulate AMPAR accumulation and spine size at basal states to maintain a neuron's ability to undergo LTP and to avoid saturating plasticity (*Araki et al., 2015*; *Zhu et al., 2002*). Here, we demonstrate that only the SynGAP-α1 isoform efficiently drives AMPAR insertion and spine enlargements during LTP (*Figure 6*). SynGAP-α1 is highly concentrated in the PSD via LLPS and PDZ-ligand-mediated interaction with PSD-95, which generates a sharp concentration gradient of SynGAP-α1 in dendritic spines that is collapsed following activity-dependent SynGAP-α1 dispersion and subsequent synaptic potentiation (*Araki et al., 2015*; *Dosemeci and Jaffe, 2010*; *Lautz et al., 2018*; *Lautz et al., 2019*; *Yang et al., 2013*; *Yang et al., 2011*). We speculate that the high magnitude of activity-dependent dispersion of SynGAP-α1 is due, in part, to the tendency of SynGAP-α1 to robustly interact with PSD-95 and to facilitate LTP-associated signaling in the synapse. Mice with *Syngap1* haploinsufficiency display exaggerated synaptic connectivity and dysregulated E/I balance in CA1 excitatory neurons (*Clement et al., 2012*). These may be a result mainly from α1-specific haploinsufficiency as SynGAP-α1 strongly regulates synaptic function compared to other isoforms.

## Distinct biochemical properties, subcellular localization patterns, GAP activity levels, and functional roles of SynGAP isoforms

We found previously that disruption of SynGAP-α1 LLPS through the LDKD mutation decreases synaptic enrichment of SynGAP-α1, and this decreases the stimulation threshold of LTP when rescued knockdown with this construct (*Zeng et al., 2016*). Our current data in *Figure 5* and our previous report (*Figure 6AB*, *Zeng et al., 2016*) provide two lines of evidence to quantify the relative contribution of phase separation and 'traditional PDZ-binding.' The lack of the PDZ ligand in SynGAP-α2 is its key difference from SynGAP-α1, The loss of the PDZ ligand decreases the affinity for PSD-95 and is shown to decrease the synaptic localization of SynGAP by ~25% (*Figure 5*). The additional lack of a complete coiled-coil domain in the SynGAP-β isoform leads to another ~25% decrease of synaptic localization. Specific mutations introduced in SynGAP-α1 to disrupt the coiled-coil-domain-dependent multimerization required for phase separation (SynGAP-α1 LDKD) and PSD-95 PDZ binding (SynGAP-α1 Δ4) led to comparable decreases in synaptic localization (*Figure 6AB*; *Zeng et al., 2016*), supporting the idea that both the direct binding to PSD-95 and the phase separation contribute to SynGAP synaptic localization. We also found in the present study that SynGAP knockdown alters dendritic development, and the α1 isoform cannot fully rescue this phenotype (*Figure 7*). However, the SynGAP-α1 LDKD was able to rescue the dendritic arborization phenotype. By specifically manipulating some intrinsic properties of SynGAP – including its localization patterns and ability to phase separate – we observe that these intrinsic properties themselves are related to distinct functional effects.

As we have shown, the SynGAP isoforms differentially regulate small G-proteins. These results may be due to differences in the localization of the SynGAP isoforms, since rates of biochemical reactions are dependent on the concentration of reactants within a microenvironment. LLPS of SynGAP physically separates the GAP domain within SynGAP from the small G proteins. This is consistent with our observation that while SynGAP-β generally showed the weakest LLPS but has the

highest GAP activity towards almost all small G-proteins (*Figure 8B*). It is known that various GAPs are also differentially localized by distinct lipid modifications. After synthesis of Ras, Rap1, and Rac1, farnesyl or geranylgeranyl moieties are attached to the C-terminal 'CAAX' motifs (C: Cys; A: an aliphatic amino acid, X: M, Q, S, T, or A for farnesyl, L or I for geranylgeranyl) for membrane tethering, facilitating interaction with effector molecules proximal to the membrane. The various small G proteins have slightly different CAAX motifs that are susceptible to distinct modifications (e.g. H-Ras contain CVIM for farnesylation, Rap1a contain CLLL for geranylgeranylation) and thus are differentially targeted to cellular microenvironments (*Moores et al., 1991*; *Simanshu et al., 2017*; *Wright and Philips, 2006*). Thus, the combinations of small G-protein localization and SynGAP isoform localization may define the ability of each SynGAP isoform to activate GTPases, and thus differentiate their function in synaptic spines or dendrites.

### Importance of characterizing various SynGAP isoforms to elucidate ID/ASD pathogenesis: Therapeutic strategies for MRD5 and neurodevelopmental disorders

*SYNGAP1* is the 4th most prevalent gene that is mutated in NDDs such as ID/ASD. Mutation of *SYNGAP1* explains ~0.75% of all NDD cases which is nearly as high as prominent X-linked disorders, such as the Fragile X syndrome (*Fitzgerald et al., 2015*). *SYNGAP1* is located in the 6p21 region, near the major immuno-histocompatibility complex (MHC) where a high rate of sequence variability is observed between people (*Reche and Reinherz, 2003*; *Sommer, 2005*). This high rate of variability within genomic regions, along with the fact that *SYNGAP1* does not have neuronal homologs despite its critical roles in development and plasticity, may contribute to the high rate of association of *SYNGAP1* with MRD5 in ID/ASD populations. We highlight the importance of SynGAP-α1 in synaptic plasticity and suggest that correcting the exaggerated downstream activity due to haploinsufficiency of SynGAP-α1 might be beneficial for patients. In this paper, we characterized dynamic changes in the expression profile of SynGAP isoforms in the brain throughout development. We showed that SynGAP-α1 expression accounts for only 25–35% of total SynGAP, highlighting the importance of assessing the function of all isoforms to understand the pathogenesis of MRD5. We also found SynGAP-β to be expressed earliest in development, and to play a unique role in dendritic arbor development. Further characterization of downstream small G proteins and kinases that have pivotal roles in dendritic development will lead to unique targets for treating the aberrant neuronal wiring that is associated with MRD5 as well as other ID/ASD-related neurodevelopmental disorders.

## Materials and methods

### Key resources table

| Reagent type (species) or resource | Designation | Source or reference | Identifiers | Additional information |
|---|---|---|---|---|
| Gene (*Rattus norvegicus*) | SynGAP-α1 | | NM_001113409.3 | |
| Gene (*Rattus norvegicus*) | SynGAP-α2 | | AF050183.2 | |
| Gene (*Rattus norvegicus*) | SynGAP-β | | AB01692.1 | |
| Gene (*Rattus norvegicus*) | SynGAP-γ | | AF058789.2 | |
| Strain, strain background *Mus musculus* | SynGAP KO mice | *Kim et al., 2003* | | Backcrossed with C57BL6 |
| Cell line (*Homo sapiens*) | HEK293T | ATCC | CRL-3216 | |

*Continued on next page*

*Continued*

| Reagent type (species) or resource | Designation | Source or reference | Identifiers | Additional information |
|---|---|---|---|---|
| Biological sample *Mus musculus* | Mouse whole brain or brain region | | | C57BL6, male and female |
| Biological sample (*Rattus norvegicus*) | Rat Hippocampal Primary Neuron | | | Days in vitro 3–21 |
| Antibody | Rabbit polyclononal antibody | *Kim et al., 1998* | JH2469 | Anti-SynGAPα1 1:1000 |
| Antibody | Rabbit polyclononal antibody | | JH7265 | Anti-SynGAPα2 1:1000 |
| Antibody | Rabbit polyclononal antibody | | JH7266 | Anti-SynGAPβ 1:1000 |
| Antibody | Rabbit polyclononal antibody | | JH7366 | Anti-SynGAPγ 1:1000 |
| Peptide, recombinant protein | SynGAPα2 C-tail | Johns Hopkins Sequencing Facility | RH376 | CPPRLQITENGEFRNTADH |
| Peptide, recombinant protein | SynGAPβ C-tail | Johns Hopkins Sequencing Facility | RH371 | CGGGGAAPGPPRHG |
| Peptide, recombinant protein | SynGAPγ C-tail | Johns Hopkins Sequencing Facility | RH377 | CRLLDAQLLIR |
| Sequence-based reagent | Primer: SG upstream sense for ScaI cloning | IDT | SJ22 | ACTGTAGCCTGG GTGTCCAATATG |
| Sequence-based reagent | Primer: alpha 2 SG reverse | IDT | SJ24 | ggattgcggccgcCTA GTGGTCTGCGGTGTTCCG |
| Sequence-based reagent | Primer: beta SG reverse | IDT | SJ25 | ggattgcggccgcTCAGC CATGGCGGGGTGGTCC |
| Sequence-based reagent | Primer: gamma SG reverse | IDT | SJ23 | ggattGCGGCCGCttacct gatgaggagCTGAGCG TCGAGCAGCCT |
| Genetic reagent (*Rattus norvegicus*) | GFP-SynGAP-α1 | *Araki et al., 2015* | | EGFP: N terminal tag |
| Genetic reagent (*Rattus norvegicus*) | GFP-SynGAP-α2 | | | EGFP: N terminal tag |
| Genetic reagent (*Rattus norvegicus*) | GFP-SynGAP-β | | | EGFP: N terminal tag |
| Genetic reagent (*Rattus norvegicus*) | GFP-SynGAP-γ | | | EGFP: N terminal tag |
| Genetic reagent (*Rattus norvegicus*) | GFP-SynGAP-α1 LDKD | *Zeng et al., 2016* | | EGFP: N terminal tag; 2-point mutations, L1202D and K1252D |

*Continued on next page*

*Continued*

| Reagent type (species) or resource | Designation | Source or reference | Identifiers | Additional information |
|---|---|---|---|---|
| Genetic reagent (*Rattus norvegicus*) | shRNA-SynGAP#5 | *Araki et al., 2015* | | shRNA sequence: CCT GGA TGA AGA CTC CAT TAT |
| Commercial assay or kit | Imject Maleimide-Activated mcKLH | Pierce | 77605 | |
| Commercial assay or kit | Pierce BCA Protein Assay Kit | Pierce | 23225 | |
| Chemical compound, drug | DL-AP5 | TOCRIS | 0105 | |
| Chemical compound, drug | Glycine | TOCRIS | 0219 | |
| Chemical compound, drug | Strychnine | SIGMA | P1675 | |
| Chemical compound, drug | Picrotoxin | TOCRIS | 1128 | |
| Chemical compound, drug | Tetrodotoxin citrate | TOCRIS | 1069 | |
| Software, algorithm | Prism 8 | GraphPad | | |
| Software, algorithm | ImageJ 1.8.0_112 | NIH | | Particle Analysis |

## Reagents and cDNA constructs

All restriction enzymes were obtained from New England Biolabs. Chemicals were obtained from SIGMA-Aldrich unless otherwise specified. TTX, Bicuculline, and Strychnine were obtained from TOCRIS Bioscience. Goat anti-SynGAP-$\alpha$1 antibody is from Santa Cruz (sc-8572). Rabbit pan-SynGAP 947–1167 antibody is from Thermo scientific (#PA-1–046). DNA sequencing was performed at the Johns Hopkins University School of Medicine Sequencing Facility. Rat SynGAP-$\alpha$1 (NCBI accession number NM_001113409.3) were cloned previously (*Kim et al., 1998*). Primers to amplify partial sequences of SynGAP-$\alpha$2, $\beta$, and $\gamma$ were designed by referring to rat SYNGAP1 genomic reference sequence NC_005119.4. These sequences were amplified by RT-PCR using rat brain total RNA as a template. These were subcloned into GFP- SynGAP-$\alpha$1 at ScaI/NotI site and $\alpha$1 specific C-terminal sequence was replaced with isoform specific sequences. HEK293T cells were obtained from ATCC (ATCC CRL-3216) and were minimized passage number in order to maintain their identity. Cells were also periodically tested the mycoplasma contamination using PCR-based MycoAlert Mycoplasma Detection Kit (Lonza #: LT07-118).

## Antibodies

The rabbit anti-SynGAP-$\alpha$1 antibody was used as described in previous reports (*Kim et al., 1998*; *Rumbaugh et al., 2006*). To raise antibodies that specifically recognize each non-$\alpha$1 SynGAP isoform, we conjugated 10–18 amino acids of the C-terminal sequences of each SynGAP isoform with an N-terminal Cysteine (CPPRLQITENGEFRNTADH (JH7265, $\alpha$2), CGGGGAAPGPPRHG (JH7266, $\beta$), and CRLLDAQLLIR (JH7366, $\gamma$)) to Keyhole limpet hemocyanin (PIERCE) using the manufacturer's protocol. Antisera acquired after 2 booster injections ($\alpha$1, $\alpha$2, $\beta$, and $\gamma$) were affinity purified using peptide coupling sulfolink-beads (PIERCE).

## Quantitative western blotting

Brain regions or organs were excised from C57BL6 mice at specified ages. Tissues were lysed in 10 volumes of lysis buffer (50 mM Tris pH 8.0, 100 mM NaCl, 1 mM EDTA, 1 mM EGTA, 1% Triton X-100, 0.2% SDS, 0.5% Sodium deoxycholate, with cOmplete Protease inhibitor EDTA-free mix (Roche/SIGMA) by Dounce A homogenizer. Protein concentrations were measured by Pierce BCA assay kit (Pierce 23225). Equal protein amounts (10 μg) were loaded into each lane. After probing by primary and secondary antibodies, signals were measured by a fluorescence-based imaging system for our quantitative western blotting (Odyssey CLx Imaging System). Fluorescence detection is suitable for quantitative immunoblotting across large dynamic ranges (*Bakkenist et al., 2015*; *Gerk, 2011*; *Wang et al., 2007*; *Weldon et al., 2008*). 50% of the first experimental lane was run in the left-most lane in order to assure the given quantification is linear in every primary-secondary antibody combination.

## Human SYNGAP1 splicing analysis

Exon junction abundance data were acquired from Brain Seq Consortium Phase 1 (*Jaffe et al., 2018*). Briefly, total RNA extracted from post-mortem tissue of the dorsolateral prefrontal cortex grey matter (DLPFC) was sequenced and reads were aligned with TopHat (v2.0.4) based on known transcripts of the Ensembl build GRCh37.67. Splice junctions were quantified by the number of supporting reads aligned by Tophat, and counts were converted to 'RP80M' values, or 'reads per 80 million mapped' using the total number of aligned reads across the autosomal and sex chromosomes (dropping reads mapping to the mitochondria chromosome), which can be interpreted as the number of reads supporting the junction in our average library size, and is equivalent to counts per million reads mapped (CPM) multiplied by 80. For a given 5' splice donor site, all identified 3' splice acceptors were grouped together to calculate the relative abundance of each splice decision.

## Characterization for biochemical properties of SynGAP isoforms

HEK 293 T cells were transfected with SynGAP and/or PSD-95 for 16 hr. Cells were lysed in 0.5 ml of assay buffer (50 mM Tris pH 8.0, 100 mM NaCl, 1 mM EDTA, 1 mM EGTA, 1% Triton X-100, 0.1% SDS, 0.5% Sodium deoxycholate, with cOmplete Protease inhibitor EDTA-free mix (Roche/SIGMA)). Lysates were centrifuged at 15000 x g for 10 min at 4°C. The supernatant-containing the soluble [S] fraction was collected. Pellets were resuspended and sonicated in 0.5 ml of assay buffer to obtain complete homogenate of pellet [P] fraction.

For imaging of LLPS dynamics in living cells, HEK 293 T cells were grown on Poly-L-Lysine-coated glass coverslips. Cells were transfected with GFP-SynGAP and/or PSD-95-mCherry for 16 hr before being placed in a custom-made live imaging chamber for observation under confocal microscopy. Cells were perfused with extracellular solution (ECS: 143 mM NaCl, 5 mM KCl, 10 mM Hepes pH 7.42, 10 mM Glucose, 2 mM CaCl$_2$, 1 mM MgCl$_2$). For DAPI staining, cells were fixed with Parafix (4% paraformaldehyde, 4% Sucrose in PBS) for 15 min at room temperature, followed by incubating with 300 nM DAPI in PBS for 5 min at room temperature. Cells were briefly washed with PBS and mounted on slideglass. Cells were observed on an LSM880 (Zeiss) microscopy with a 40x objective lens (NA 1.3).

## Cellular localization assay of SynGAP isoforms in HEK 293 T cells

HEK 293 T cells plated on 18 mm coverslips coated with poly-L-lysine were transfected using LipofectAMINE 2000 for 16 hr. Cells were fixed with 4% paraformaldehyde and sucrose in PBS for 15 min and coverslips were mounted on slideglass. After taking images of 4 randomly selected regions containing >30 cells from each coverslip (with the experimenter blind to the transfection conditions) using an LSM 880 confocal microscope, the percentages of cells with PSD-95 puncta (>1 μm diameter) were measured by puncta analysis function in Image J software. The averages and SEM of all 4 regions were calculated and displayed in graph.

## PSD fractionation

Fractionation of post-synaptic density (PSD) was performed as previously described (*Kohmura et al., 1998*). In brief, mouse brains were collected and homogenized by 10–15 strokes of a Dounce A homogenizer in Buffer A (0.32M Sucrose, 10 mM Hepes (pH7.4) with cOmplete protease inhibitor

mix (SIGMA)). The homogenate was centrifuged at 1000 x g for 10 min at 4°C. The supernatant (Post Nuclear Supernatant; PNS) was collected and centrifuged at 13,800 x g for 20 min at 4°C. The pellet (P2 fraction) was re-homogenized in 3 volumes of Buffer A. The re-homogenized P2 fraction was layered onto a discontinuous gradient of 0.85, 1.0, 1.2 M sucrose (all containing 10 mM Hepes (pH7.4) plus cOmplete protease inhibitor mix), and were centrifuged at 82,500 x g for 2 hr at 4°C (Beckman SW28 swing rotor). The band between 1.0 and 1.2 M sucrose was collected as the synaptosome fraction and diluted with 80 mM Tris-HCl (pH 8.0). An equal volume of 1% Triton X-100 was added and rotated for 15 min at 4°C followed by centrifuging 32,000 x g for 20 min. The supernatant was collected as a Triton-soluble synaptosome (Syn/Tx) fraction, and the pellet was re-homogenized in Buffer A by applying 10 passes through a 21G syringe. Equal amounts of protein (10 µg for immunoblotting) were used for further assay.

## Chemical LTP stimulation and quantification

Live imaging and quantification of LTP were performed as described previously (*Araki et al., 2015*). Hippocampal neurons from embryonic day 18 (E18) rats were seeded on 25 mm poly-L-lysine-coated coverslips. The cells were plated in Neurobasal media (Gibco) containing 50 U/ml penicillin, 50 mg/ml streptomycin and 2 mM GlutaMax supplemented with 2% B27 (Gibco) and 5% horse serum (Hyclone). At DIV 6, cells were thereafter maintained in glia-conditioned NM1 (Neurobasal media with 2 mM GlutaMax, 1% FBS, 2% B27, 1 x FDU (5 mM Uridine (SIGMA F0503), 5 mM 5-Fluro-2'-deoxyuridine (SIGMA U3003). Cells were transfected at DIV17-19 with Lipofectamine 2000 (Invitrogen) in accordance with the manufacturer's manual. After 2 days, coverslips were placed on a custom perfusion chamber with basal ECS (143 mM NaCl, 5 mM KCl, 10 mM Hepes pH 7.42, 10 mM Glucose, 2 mM CaCl$_2$, 1 mM MgCl$_2$, 0.5 µM TTX, 1 µM Strychnine, 20 µM Bicuculline), and time-lapse images were acquired with either LSM880 (Carl Zeiss) or Spinning disk confocal microscopes controlled by Axiovision software (Carl Zeiss). Following 5–10 min of baseline recording, cells were perfused with 10 ml of glycine/0 Mg ECS (143 mM NaCl, 5 mM KCl, 10 mM HEPES pH 7.42, 10 mM Glucose, 2 mM CaCl$_2$, 0 mM MgCl$_2$, 0.5 µM TTX, 1 µM Strychnine, 20 µM Bicuculline, 200 µM Glycine) for 10 min, followed by 10 ml of basal ECS. To stabilize the imaging focal plane for long-term experiments, we employed Definite focus (Zeiss). For quantification, we selected pyramidal neurons based on morphology that consisted of a clear primary dendrite, and quantified all spines on the 30–40 µm stretch of the secondary dendrite beginning just after the branch from the primary dendrite. For identifying spine regions, we used the mCherry channel to select the spine region that was well separated from dendritic shaft. These regions of interest (ROIs) in the mCherry channel were transferred to the green channel to quantify total SynGAP content in spines. Total spine volume was calculated as follows; (Average Red signal at ROI – Average Red signal at Background region) * (Area of ROI). Total SynGAP content was calculated as follows; (Average Green signal at ROI – Average Green signal at Background region) * (Area of ROI). Through this quantification, we can precisely quantify the total signals at each spine even if the circled region contained some background area. For [%] spine enlargement before/after LTP, we took a relative ratio of these total spine volume (total red signal) of each spine before/after LTP ([%] spine enlargement = (Total Red Signal after chemLTP/Total Red signal at basal state-1)*100). For [%] SynGAP dispersion, we calculated the degree of total SynGAP content loss after chemLTP at each spine compared to the total SynGAP content at basal state ([%] dispersion = (1- Total Green Signal after chemLTP/Total Green signal at basal state) * 100).

## Fluorescence recovery after photobleaching (FRAP) of single dendritic spines

Rat hippocampal neurons were prepared as in 'Chemical LTP stimulation and quantification.' Neurons were transfected at DIV 17–19 with Lipofectamine 2000 (Invitrogen), and the experiment was performed 36–48 hr following transfection. Neurons were imaged using an LSM 880 confocal microscope using a custom-made live-cell imaging chamber filled with basal ECS at 37°C. Short stretches of dendrites of pyramidal neurons were imaged using a 63X objective, and only one dendritic segment was imaged per each 1 hr imaging session. Confocal Z-stack images were acquired every 60 s using 488 nm and 563 nm lasers for excitation of GFP (SynGAP) and mCherry (shRNA reporter and cell-fill), respectively. Two baseline Z-stacks were acquired before photobleaching with the 488 nm

laser (100% intensity, 25 iterations) at multiple ROIs drawn around single dendritic spines by the experimenter. Z-stack images were acquired every 60 s until the end of the 60 min experiment. Photobleaching laser intensity and image acquisition parameters were kept constant across experiments. Images were analyzed using ImageJ. A median filter was applied to the maximum intensity projection of each channel-split time course image before rigid body registration using the plugin Multi-StackReg. Mean intensity values across all timepoints were extracted from each FRAP ROI, two additional non-bleach ROIs to be used for correcting for photobleaching during image acquisition, and two background ROIs (one above and one below the horizontal dendritic segment) for subtraction of background signal. All values were normalized to bleach depth in order to extract the recovery fraction. Nonlinear regression was performed on each dataset, and the data were fit using first-order exponentials using GraphPad Prism 8 software. Curve plateaus were statistically compared using the extra sum-of-squares F test.

## Dendritic Arbor development assay

Cultured hippocampal neurons were plated on coverslips as described above and were co-transfected at DIV 3–4 with pSUPER-SynGAP shRNA and shRNA-resistant GFP-SynGAP-$\alpha$1, $\alpha$2, $\beta$, and $\gamma$ replacement constructs. pCAG-DsRed2 was also co-transfected as a cell-fill for morphological analysis. Neurons were fixed at DIV 8–9 by incubating them with Parafix (4% paraformaldehyde, 4% Sucrose in PBS) for 15 min at room temperature, followed by incubation with 300 nM DAPI in PBS for 5 min at room temperature. Cells were briefly washed with PBS and mounted onto glass slides. Cells were imaged with a LSM880 (Zeiss) confocal microscope equipped with a 40x objective lens (NA 1.3) and GaAsP detectors. To obtain Sholl profiles of dendritic arbors, images of entire dendritic arbors of hippocampal neurons expressing DsRed were acquired and processed using Image J (Fiji) software. Scholl analysis consisted of drawing concentric rings with radii of 10, 20, 30, 40, 50, 100, and 150 µm from the center of the cell body and counting the number of dendritic intersections across each concentric circle. If a branch point fell on a line, it was counted as two crossings (*Nakayama et al., 2000*).

## Small GTPase activity assay

Small GTPase activity was measured using a small GTPase-GTP pull-down assay. DNA constructs expressing a small G protein and a single SynGAP isoform were co-transfected into HEK 293 T cells for 48–72 hr. Active Ras levels were then assayed using a Ras activation assay kit (EMD Millipore). In brief, cells were lysed in $Mg^{2+}$ lysis/wash buffer (25 mM HEPES pH 7.5, 150 mM NaCl, 1% Igepal CA-630, 10 mM $MgCl_2$, 1 mM EDTA, 10% glycerol), and active GTP-bound small G-proteins were pulled down using beads covalently bound to effector domains. After washing beads, active GTP-bound small G proteins were recovered through the addition of 2x SDS sample buffer followed by SDS-PAGE and subsequent immunoblotting for the various small G proteins.

## Statistics

All data are expressed as means ± S.E.M. of values unless otherwise stated. One-way ANOVAs were used, followed by Tukey post hoc for multiple comparisons unless otherwise specified. If the interaction between two-factors was observed by two-way ANOVA, we performed individual Tukey *post hoc* tests to compare the measures as a function of one factor in each fixed levels of another factor unless otherwise specified. Statistical analyses and preparations of graphs were performed using SPSS 9.0, Excel 2010, or GraphPad Prism 4.0/5.0/8.0 software (*p<0.05; **p<0.01; ***p<0.001).

## Acknowledgements

We thank all members of the Huganir lab for discussion and support throughout this work especially Drs. Kacey Rajkovich, Elizabeth Gerber, Megnan Tian, Bian Liu, and Rich Johnson for critical experimental help and preparation of the manuscript. We also thank Andrew E Jaffe, and Daniel R Weinberger for help with acquiring and analyzing of RNAseq data. This work was supported by grants from National Institute of Health (MH112151, NS036715) and the SynGAP Research Fund. We want to thank the Bridge The Gap SYNGAP Education and Research Foundation, the SynGAP Research Fund, and all of the SynGAP patient families for their outreach and advocacy.

# Additional information

## Funding

| Funder | Grant reference number | Author |
| --- | --- | --- |
| National Institutes of Health | MH112151 | Richard L Huganir |
| National Institutes of Health | NS036715 | Richard L Huganir |

The funders had no role in study design, data collection and interpretation, or the decision to submit the work for publication.

## Author contributions

Yoichi Araki, Conceptualization, Data curation, Formal analysis, Validation, Visualization, Writing - original draft, Writing - review and editing; Ingie Hong, Formal analysis, Investigation, Writing - review and editing; Timothy R Gamache, Investigation, Writing - original draft, Writing - review and editing; Shaowen Ju, Resources, Investigation; Leonardo Collado-Torres, Joo Heon Shin, Formal analysis, Investigation; Richard L Huganir, Conceptualization, Supervision, Funding acquisition, Project administration, Writing - review and editing

## Author ORCIDs

Yoichi Araki (iD) https://orcid.org/0000-0002-3455-9377
Ingie Hong (iD) https://orcid.org/0000-0002-7246-9233
Timothy R Gamache (iD) https://orcid.org/0000-0002-7357-2857
Shaowen Ju (iD) https://orcid.org/0000-0002-1365-9803
Leonardo Collado-Torres (iD) http://orcid.org/0000-0003-2140-308X
Richard L Huganir (iD) https://orcid.org/0000-0001-9783-5183

## Ethics

Animal experimentation: This study was performed in strict accordance with the recommendations in the Guide for the Care and Use of Laboratory Animals of the National Institutes of Health. All of the animals were handled according to approved institutional animal care and use committee (IACUC) protocol # MO20M92 of Johns Hopkins University School of Medicine. The protocol was approved by the Animal Care and Use Committee at Johns Hopkins University School of Medicine.

## Decision letter and Author response

Decision letter https://doi.org/10.7554/eLife.56273.sa1
Author response https://doi.org/10.7554/eLife.56273.sa2

# Additional files

## Supplementary files
- Transparent reporting form

## Data availability

All data generated or analysed during this study are included in the manuscript and supporting files.

The following previously published dataset was used:

| Author(s) | Year | Dataset title | Dataset URL | Database and Identifier |
| --- | --- | --- | --- | --- |
| Jaffe AE, Straub RE, Shin JH, Tao R, Gao Y, Collado-Torres L, Kam-Thong T, Xi HS, Quan J, Chen Q, | 2018 | BrainSeq Phase1 DLPFC | http://doi.org/10.7303/syn12299750 | Synapse, 10.7303/syn12299750 |

Colantuoni C, Ulrich WS, Maher BJ, Deep-Soboslay A, BrainSeq Consortium, Cross AJ, Brandon NJ, Leek JT, Hyde TM, Kleinman JE, Weinberger DR

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
