## [Decision Letter]

**Acceptance summary:**

In this paper, Araki et al. systematically surveyed expression and functions of the family of SynGAP splice isoforms. This is important work, as the molecule underlies relatively common and variably severe intellectual disability, is a prominent synaptic component, and exerts strong control over synaptic structure and function. The results provide critical information about isoform-specific developmental time course, breaks new ground in ascribing control of dendrite growth patterns to different isoforms, and links functional outcomes to particular domains of the molecule which control subcellular localization and thus availability for GAP activity.

**Decision letter after peer review:**

Thank you for submitting your article "Distinct phase separation properties of SynGAP isoforms regulate synaptic plasticity and dendritic development" for consideration by *eLife*. Your article has been reviewed by three peer reviewers, and the evaluation has been overseen by Gary Westbrook as the Senior Editor. The following individual involved in review of your submission has agreed to reveal their identity: Thomas Blanpied (Reviewer #1).

The reviewers have discussed the reviews with one another and the Senior Editor has drafted this decision to help you prepare a revised submission. We would like to draw your attention to changes in our revision policy that we have made in response to COVID-19 (https://elifesciences.org/articles/57162). The below revisions are considered essential and in line with *eLife*'s new revision policy. The original reviews are also provided and we ask that you include a point-by-point response to the reviews, which are largely summarized in the essential revisions paragraph, but also include some straightforward points to address.

Essential revisions

1) A more balanced interpretation of the results is necessary without the tendency to overinterpret as noted by all reviewers. We think this will require a significant re-write of the manuscript while also considering point 2 below.

2) The experiments are not convincing regarding the phase separation properties of the specific isoforms. We strongly recommend a refocus of the manuscript on isoform specific functions. These points are clearly stated in reviewer 2's comments and endorsed in our post-review discussion.

3) With respect to the second comment of reviewer 1 and the second comment of reviewer 3 in the original reviews, these would seemingly require additional experiments, which unless the authors have the data in hand, *eLife* is not requiring under the current circumstances. If you have the data, please add. Otherwise please address the issue by adjusting your interpretations in the text.

Reviewer #1:

Araki et al. systematically survey expression and functions of the family of SynGAP splice isoforms. This is important work, as the molecule underlies relatively common and variably severe intellectual disability, is a prominent synaptic component, and has been shown by these authors and others to exert strong control over synaptic structure and function. The work provides critical information about isoform-specific developmental time course, breaks new ground in ascribing control of dendrite growth patterns to different isoforms, and links functional outcomes to particular domains of the molecule which control subcellular localization and thus availability for GAP activity. Technically, the work is sound and the document is clearly written.

Addressing a few concerns would clarify some remaining issues and provide tighter support for some key conclusions.

1) The authors emphasize repeatedly that the developmental expression profile is unique for α1 vs. α2 or β. This is a very attractive simplification, but Figure 1G demonstrate striking similarities rather than differences in the expression timeline, and the impressive accumulation of human RNA seq data in Figure 1I and J show little developmental changes at all in the ratio of transcript levels from 9 wks gestation to 90 years of age. In mouse, β does start to decrease around the time α peaks, but I don't see how the magnitude of these changes can be emphasized so strongly as a mechanism underlying the human disorders stemming from SynGAP haploinsufficiency.

2) The unique functional profile of β is ascribed to its lack of LLPS propensity and thus lack of synapse targeting. This could be tested directly by simply adding an exogenous synaptic targeting signal (PDZ-binding or other). I would guess that it would be fairly straightforward to test the prediction that this construct would fail to rescue dendrite growth in the knockdown; the LTP-related predictions also would be interesting to test but seem much more intensive and the effort may not be warranted.

3) The authors unequivocally ascribe all synaptic accumulation (and consequent function) to LLPS. Again, this is an attractive conclusion, but the direct evidence for LLPS vs. "traditional" binding within the PSD is limited, is it not? It seems difficult here to ascribe a particular function to LLPS per se. If phase separation in synapses and accumulation without certain particular features of phase separation itself are essentially indistinguishable experimentally, this should be acknowledged. The PSD is after all a highly ordered megamolecular structure, and the respective roles of discrete protein-protein interactions vs phase separation in creating this order still seem unknown.

Reviewer #2:

This study compares roles of the four different SynGAP C-terminal splice isoforms (α1, α2, β and γ) in synaptic function and development. The authors report that heterologous expression of individual SynGAP isoforms present (1) different GAP activity and ability to recruit PSD95 in HEK cells, (2) different expression levels at synapses, (3) different dispersion kinetics from dendritic spines upon induction of chemical LTP, and (4) different ability to rescue either neuronal morphology or chemical LTP-induced spine enlargement and AMPA receptor insertion. The authors further perform biochemical experiments to assess whether and how the different isoforms contribute to the generation of phase condensates together with PSD-95. There is value in a systematic analysis of the localization and function of the various SynGAP isoforms. However, the paper overstates some of the at best correlative findings, and overall this paper is borderline for *eLife* in its current form in my view.

Given the current coronavirus crisis, I am not suggesting further experimentation. Instead, addressing the following points in textual revision and data analyses, the paper may become suitable for *eLife*.

Key concern:

The authors use two measurements for phase separation that, together, are not sufficient to establish phase separation. First, the sedimentation-based assay performed in cell lysates (Figures 2A-2C) cannot distinguish between phase condensates and aggregates, both would end up in the pellet (this is different from its typical use in which the experiment starts with purified protein in solution and aggregates are removed before the experiment starts). Second, in the transfection assay (Figures 2D, 2E), all SynGAP isoforms form puncta of variable size, but only the α1-isoform forms puncta that colocalize with co-transfected PSD-95. Although phase separation results in puncta, not all puncta reflect phase transitions, and in this experiment it is impossible to tell whether only SynGAP-α1+PSD-95 from phase condensates, or whether all SynGAP isoforms form them but PSD-95 is only recruited into the condensates when there is direct binding to PSD-95 (which would be different from the initial findings described in Zeng et al., 2016, but match better with the data shown in Figure 2A-2C). Ultimately, the conclusion that can be drawn is the recruitment of PSD-95 by individual SynGAP isoforms, and the experiment cannot strongly speak to LLPS (without, for example, using FRAP of the puncta to establish turnover rates as a key distinguishing property between phase condensates and aggregates).

Hence, the current data do not provide strong evidence for phase separation properties of the various SynGAP isoforms. At the very least, the authors would have to remove the causative claims of the relationship between phase transition and SynGAP function (most importantly from the title, but careful phrasing should be used everywhere), re-write the Results section and relabel Figures 2A-2C such that it is not assumed that everything that is in the pellet is a phase condensate. The model of phase separation should be appropriately discussed in the Discussion section, but results etc. should not assume that the experiments establish phase transitions because they do not. This would lead to a refocus of the paper on the isoform-specific functions of SynGAP, which is really what the data provide insight in, and in my view would be suitable for *eLife*.

Reviewer #3:

In this manuscript, Araki et al. investigate the expression and the role for the different isoforms of the gene SYNGAP1, whose many loss of function mutations were associated with various neurodevelopmental diseases. Multiple outcomes for the mutations in the same gene make the study of SYNGAP1 isoforms essential for a better understanding of the disease origin.

The manuscript is well written and addresses a topic of likely broad interest. Overall, the experiments are well designed and use a nice combination of techniques, different tools are validated with correct controls, and the data is solid and followed up by reasonable conclusions.

However, there are two issues which should be addressed before this work is suitable for publication in *eLife*.

1) According to the Materials and methods, normality of the samples seems to not be tested, however, ANOVA tests are parametric and assume normality of the data tested. Authors should address this issue and if necessary, perform the appropriate statistical analysis.

2) Differential regulation of GTPase activity by the different isoforms is an important topic of the manuscript, but despite the strong and specific tools designed by the authors, they perform their assay only in cell line. However, authors show a specific isoform localization in neurons in compartments which don't exist in HEK cells. Furthermore, they show that neuronal protein partner PSD95 has a dramatic effect of the localization. Authors should address this issue by ICC and western blot with one experiment in neurons.

---

## [Author Response]

Essential revisions1) A more balanced interpretation of the results is necessary without the tendency to overinterpret as noted by all reviewers. We think this will require a significant re-write of the manuscript while also considering point 2 below.

We thank the Senior editor and all reviewers for the positive and constructive comments. We have now significantly rewritten the manuscript and changed the title based on the comments.

2) The experiments are not convincing regarding the phase separation properties of the specific isoforms. We strongly recommend a refocus of the manuscript on isoform specific functions. These points are clearly stated in reviewer 2's comments and endorsed in our post-review discussion.

As per these recommendations, we rewrote the manuscript and are now placing focus on the isoform-specific functions of Syngap1 rather than the phase separation properties.

We respectfully point out that we have validated our sedimentation-based assay in Figure 2B, employing a phase separation mutant form of SynGAP (α1 LDKD) which was extensively characterized in an earlier publication (Zeng et al., 2016). The large differential between the pellet sedimentation of α1 LDKD+PSD95 (5th column of graph) compared to α1 WT+PSD95 (3rd column of graph) shows that the assay can discriminate bona fide phase-separation fraction from non-specific aggregation.

We have also added new data (new Figure 2—figure supplement 1A) showing that our GFP-tagged full-length version SynGAP-α1 LDKD exhibits enhanced fluorescence recovery after photobleaching (FRAP) of dendritic spine spines, suggesting that disruption of SynGAP LLPS without disrupting PDZ-binding leads to enhanced synaptic mobility of SynGAP, likely due to a decrease in PSD association resulting from diminished LLPS propensity. In other words, these data suggest that LLPS – in addition to PDZ-domain-binding – is crucial for robust synaptic enrichment and clustering of SynGAP at the PSD. These data are consistent with previous findings showing that the LDKD mutation decreases the synaptic enrichment of SynGAP in cultured neurons (Zeng et al., 2016).

However, we understand that we cannot fully answer reviewer 2's key concern without additional data (e.g. FRAP data on SynGAP isoforms in live heterologous cells or purified proteins in vitro). Thus, we changed the title and carefully revised the manuscript based on the reviewer's comment.

3) With respect to the second comment of reviewer 1 and the second comment of reviewer 3 in the original reviews, these would seemingly require additional experiments, which unless the authors have the data in hand, eLife is not requiring under the current circumstances. If you have the data, please add. Otherwise please address the issue by adjusting your interpretations in the text.

Thank you very much for understanding our current situation. Johns Hopkins University and our laboratory is currently shut down and is fully compliant with COVID-19 related regulations. We currently do not have FRAP data on all of the isoforms. We therefore have carefully discussed the current limitations of our data in order to address the reviewers’ concerns.

Reviewer #1:1) The authors emphasize repeatedly that the developmental expression profile is unique for α1 vs. α2 or β. This is a very attractive simplification, but Figure 1G demonstrate striking similarities rather than differences in the expression timeline, and the impressive accumulation of human RNA seq data in Figure 1I and J show little developmental changes at all in the ratio of transcript levels from 9 wks gestation to 90 years of age. In mouse, β does start to decrease around the time α peaks, but I don't see how the magnitude of these changes can be emphasized so strongly as a mechanism underlying the human disorders stemming from SynGAP haploinsufficiency.

We thank the reviewer #1 for his positive and constructive comments. We have revised the manuscript to decrease the emphasis of the developmental time course of the isoforms.

2) The unique functional profile of β is ascribed to its lack of LLPS propensity and thus lack of synapse targeting. This could be tested directly by simply adding an exogenous synaptic targeting signal (PDZ-binding or other). I would guess that it would be fairly straightforward to test the prediction that this construct would fail to rescue dendrite growth in the knockdown; the LTP-related predictions also would be interesting to test but seem much more intensive and the effort may not be warranted.

Thank you for the constructive comment. As discussed in the Senior editor’s summary, we are currently not able to start new experiments because of the COVID-19 pandemic. Instead, we will discuss this concern. Thank you again for the suggestion.

3) The authors unequivocally ascribe all synaptic accumulation (and consequent function) to LLPS. Again, this is an attractive conclusion, but the direct evidence for LLPS vs. "traditional" binding within the PSD is limited, is it not? It seems difficult here to ascribe a particular function to LLPS per se. If phase separation in synapses and accumulation without certain particular features of phase separation itself are essentially indistinguishable experimentally, this should be acknowledged. The PSD is after all a highly ordered megamolecular structure, and the respective roles of discrete protein-protein interactions vs. phase separation in creating this order still seem unknown.

We agree that our claim that all synaptic accumulation (and consequent function) to LLPS is too strong and that distinguishing the individual contributions of each molecular interaction in the large PSD structure is challenging. This is a general problem in the LLPS field as phase transitions are more easily studied in vitro in simple systems and it is hard to extrapolate to more complex systems in cells. Our current data in Figure 4 and our previous report (Zeng et al., 2016, Figure 6A,B) provide two lines of evidence to quantify the relative contribution of phase separation and “traditional binding”. The lack of the PDZ ligand in SynGAP-α2 is the key difference with the SynGAP-α1 isoform, which disrupts the direct affinity towards PSD-95 and is shown to decrease the synaptic localization of SynGAP about 25% (Figure 4). The additional lack of a complete coiled-coil domain in the SynGAP-β isoform, which is critical for phase separation, leads to another ~25% decrease of synaptic localization. Specific mutations introduced in SynGAP-α1 to disrupt phase separation (L-D and K-D) and PSD-95 PDZ binding (∆4) led to comparable decreases in synaptic localization (Zeng et al., 2016, Figure 6A,B), supporting the idea that both the direct binding to PSD-95 and the phase separation contribute to SynGAP synaptic localization. Of note, these results likely underestimate the contribution because despite of the shRNA-induced knockdown, the remaining endogenous SynGAP likely limits the impact of these mutations through interaction with the exogenously expressed SynGAP. We have revised the text and discussion to clearly reflect these findings and the caveats of our conclusions.

It is true we can only see the "correlations" of phase separation deficient mutants (LDKD mutant) with (i) synaptic localization, (ii) plasticity function, and (iii) role of dendritic arborization. We already observed phase-separation mutation of SynGAP-α1 resulted in (i) deficits in synaptic location of SynGAP (Zeng et al., 2016, Figure 6A,B), (ii) deficits in synaptic plasticity regulation (i.e. threshold of LTP is decreased) (Zeng et al., 2016, Figure 7C,D), and (iii) better rescue in dendritic arborization upon SynGAP knockdown (this paper, Figure 6). Although these "correlations" are clear, we understand we cannot ascribe all these functions were explained by LLPS. We have revised the Discussion of this point in the manuscript.

Reviewer #2:Key concern:The authors use two measurements for phase separation that, together, are not sufficient to establish phase separation. First, the sedimentation-based assay performed in cell lysates (Figures 2A-2C) cannot distinguish between phase condensates and aggregates, both would end up in the pellet (this is different from its typical use in which the experiment starts with purified protein in solution and aggregates are removed before the experiment starts). Second, in the transfection assay (Figures 2D, 2E), all SynGAP isoforms form puncta of variable size, but only the α1-isoform forms puncta that colocalize with co-transfected PSD-95. Although phase separation results in puncta, not all puncta reflect phase transitions, and in this experiment it is impossible to tell whether only SynGAP-α1+PSD-95 from phase condensates, or whether all SynGAP isoforms form them but PSD-95 is only recruited into the condensates when there is direct binding to PSD-95 (which would be different from the initial findings described in Zeng et al., 2016, but match better with the data shown in 2A-2C). Ultimately, the conclusion that can be drawn is the recruitment of PSD-95 by individual SynGAP isoforms, and the experiment cannot strongly speak to LLPS (without, for example, using FRAP of the puncta to establish turnover rates as a key distinguishing property between phase condensates and aggregates).Hence, the current data do not provide strong evidence for phase separation properties of the various SynGAP isoforms. At the very least, the authors would have to remove the causative claims of the relationship between phase transition and SynGAP function (most importantly from the title, but careful phrasing should be used everywhere), re-write the Results section and relabel Figures in 2A-2C such that it is not assumed that everything that is in the pellet is a phase condensate. The model of phase separation should be appropriately discussed in the Discussion section, but results etc should not assume that the experiments establish phase transitions because they do not. This would lead to a refocus of the paper on the isoform-specific functions of SynGAP, which is really what the data provide insight in, and in my view would be suitable for eLife.

We thank the reviewer #2 for the constructive comments and have addressed them above (Essential revisions #2).

Reviewer #3:1) According to the Materials and methods, normality of the samples seems to not be tested, however, ANOVA tests are parametric and assume normality of the data tested. Authors should address this issue and if necessary, perform the appropriate statistical analysis.

We thank the reviewer #3 for the constructive comments. We retested the normality and distribution of our data. We added the detailed description regarding this matter and used non-parametric tests where appropriate.

2) Differential regulation of GTPase activity by the different isoforms is an important topic of the manuscript, but despite the strong and specific tools designed by the authors, they perform their assay only in cell line. However, authors show a specific isoform localization in neurons in compartments which don't exist in HEK cells. Furthermore, they show that neuronal protein partner PSD95 has a dramatic effect of the localization. Authors should address this issue by ICC and western blot with one experiment in neurons.

As discussed in the Senior editor’s summary, we are not able to restart new experiments because of the COVID-19 pandemic. Instead, we revised the Discussion to address this concern. Thank you for the thoughtful and constructive comment.